# Deep convective influence on the UTLS composition in the Asian Monsoon Anticyclone region: 2017 StratoClim campaign results

Silvia Bucci[1], Bernard Legras[1], Pasquale Sellitto[2], Francesco D'Amato[3], Silvia Viciani[3], Alessio Montori[3], Antonio Chiarugi[4], Fabrizio Ravegnani[5], Alexey Ulanovsky[6], Francesco Cairo[5], and Fred Stroh[7]

[1]Laboratoire de Météorologie Dynamique (LMD), UMR 8539, CNRS, École Normale Supérieure, PSL Research University, École Polytechnique, Sorbonne Université, École des Ponts ParisTech, Institut Pierre Simon Laplace, Paris, France
[2]Laboratoire Inter-universitaire des Systèmes Atmosphériques (LISA), UMR 7583, CNRS, Universitè Paris-Est-Créteil, Université de Paris, Institut Pierre-Simon Laplace, Créteil, France
[3]National Institute of Optics (CNR-INO), Firenze and Sesto Fiorentino, Italy
[4]National Institute of Geophysics and Vulcanology (INGV), Pisa, Italy (present address: SENSIT Technologies, Valparaiso, Indiana, USA)
[5]Institute for Atmospheric Sciences and Climate of the National Research Council, (ISAC-CNR), Bologna and Rome, Italy
[6]Central Aerological Observatory (CAO), Moskow, Russia
[7]Institute of Energy and Climate Research, Stratosphere, Forschungszentrum Jülich, Jülich, Germany

*Correspondence to:* Silvia Bucci (sbucci@lmd.ens.fr), Bernard Legras (bernard.legras@lmd.ens.fr)

**Abstract.** The StratoClim stratospheric aircraft campaign took place in summer 2017 in Nepal (the 27th of July - 10th of August) and provided for the first time a wide dataset of observations of air composition inside the Asian Monsoon Anticyclone. In the framework of this project, with the purpose of modelling the injection of pollutants and natural compounds into the stratosphere, we performed a series of diffusive back-trajectories runs along the flights' tracks. The availability of in-situ measurements of trace gases has been exploited to evaluate the capability of the trajectory system to reproduce the transport in the Upper Troposphere - Lower Stratosphere (UTLS) region. The diagnostics of the convective sources and mixing in the air parcel samples have been derived by combining the trajectories output with high-resolution observations of cloud tops from the Meteosat Second Generation (MSG1) and Himawari geostationary satellites. Back-trajectories have been calculated using meteorological fields from European Centre for Medium-Range Weather Forecasts (ECMWF) Reanalysis (ERA-Interim and ERA-5) at 3h and 1h resolution, using both kinematic and diabatic vertical motion. The comparison of the different trajectory runs shows, in general, a higher consistency with observed data, as well as a better agreement between the diabatic and kinematic version, when using ERA-5 based runs rather than ERA-Interim. Overall, a better capacity in reproducing the pollution features is finally found in the diabatic version of the ERA-5 runs. We adopt therefore this setting to analyse the convective influence in the UTLS starting from the StratoClim observations. A large variety of transport conditions have been individuated during the 8 flights of the campaign. The larger influence by convective injections is from the continental sources of China and India. Only a small contribution appears to be originated from maritime regions, in particular the South Pacific and the Bay of Bengal which was not particularly active during the period of the campaign, taking place during a break phase of the monsoon. However, a mass of clean air injected from a typhoon has been detected at around 18 km. Thin filamentary structures of polluted air, characterized by peaks in CO, has been observed, mostly associated with young convective air (age

less than a few days) and a predominant South-China origin. The analysis revealed a case of direct injection of highly polluted air close to the level of the tropopause (anomalies of around 80 ppbv injected at 16 km) that kept raising inside the anticyclonic circulation. Due to the location of the campaign, air from continental India, on the contrary, has been only observed to be linked to air masses that recirculated within the anticyclone for 10 to 20 days, resulting in a lower concentration of the trace gas. The analysis of a flight overpassing an intense convective system close to the Nepalese South borders, revealed the injection of very young air (few hours of age) directly in the tropopause region ($\sim$18 km), visible in the trace gases as an enhancement in CO and a depletion in $O_3$. From the whole campaign, a vertical stratification in the age of air has been observed: up to 15 km, the age is less than 3 days and these fresh air masses constitute almost the totality of the air composition. A transition layer is then individuated between 15 km and 17 km, where the convective contribution is still dominant and the ages vary between one and two weeks. Above this level, the mean age of the air sampled by the aircraft is estimated to be larger than 20 days. There, the convective contribution rapidly decreases with height, and finally became negligible around 20 km.

## 1 Introduction

The UTLS dynamics of the Northern Hemisphere (NH) during the summer season (June to August) is dominated by the Asian summer Monsoon Anticylone (AMA) system (Randel and Park, 2006). In the AMA regions, air masses from the Boundary Layer (BL) can be effectively uplifted to the UTLS and subsequently transported to the stratosphere, with a vertical transport resulting from the interaction of deep convection with the strong anticyclonic flow (Randel et al., 2010; Lawrence, 2011). Deep convection can also directly transport air close to the stratosphere during intense overshooting events. Once transported into the UTLS, pollutants remain partly trapped inside the Asian Monsoon Anticyclone (Randel et al., 2010; Vernier et al., 2011). High concentrations of tropospheric trace gases as carbon monoxide (CO), nitrogen oxides (NOx), peroxyacetyl nitrate (PAN) and hydrogen cyanide (HCN) were detected by satellites measurements inside the anticyclone borders (Park, 2004; Randel et al., 2010; Santee et al., 2017), while low concentrations of stratospheric tracers were observed (Park et al., 2008; Konopka et al., 2010). Similarly, the AMA also appears as a region of enhanced concentration of water vapour (Forster and Shine, 2002; Dvortsov and Solomon, 2001; Santee et al., 2017), related to the air mass transport from the upper troposphere (Rolf et al., 2018; Nützel et al., 2019). This confinement may favor chemical and microphysical transformations inside the AMA and may significantly affect the radiative balance, and therefore climate, at regional to global scale (Solomon et al., 2010; Sherwood et al., 2010). An additional effect of the convective uplift in the AMA is the formation of the Asian Tropopause Aerosol Layer (ATAL, Vernier et al. (2011)), extending between 13 and 18 km over Asia. This layer has been proved to exert a shortwave direct radiative forcing at the top of the atmosphere with values between -0.1 $\mathrm{Wm^{-2}}$ (Vernier et al., 2015) and -0.05 $\mathrm{Wm^{-2}}$ (Kloss et al., 2019). While it has been demonstrated that the AMA clearly provides an effective pathway for trapping tropospheric pollutants and water vapour in the atmosphere (Ploeger et al., 2013; Vogel et al., 2015), the mechanisms of transport and the distribution of the sources are still not well known (Vogel et al., 2012). A study from Chen et al. (2012), based on Lagrangian model simulations driven by Global Forecast System (GFS) winds suggested that, during summer season, the BL to the Tropical Tropopause Layer (TTL) transport in the whole Asian monsoon regions is dominated by the Western

Pacific region and the South China Seas. Other relevant sources would include, in order, the Bay of Bengal and South Asian subcontinent and, to a lesser extent, the Tibetan Plateau. Bergman et al. (2013), with a similar Lagrangian approach based on MERRA winds, indicated that, at 100 hPa inside the AMA, the main contribution comes from the Tibetan Plateau and India to a similar extent ($\sim 40\%$ and $\sim 30\%$ respectively) while the West Pacific is contributing just $\sim 10\%$. Results from Vogel et al. (2015), based on the analysis of artificial emission tracers from a chemical Lagrangian model for summer 2012, confirmed that the Northern Indian subcontinent (with the Tibetan Plateau), and Southeast Asia including the eastern part of the Bay of Bengal, represent the most important source regions for the chemical composition of the AMA at the tropopause level. Tissier and Legras (2016), with a study based on backward and forward trajectories on ERA-Interim winds between the top of convective clouds to the tropopause, found that, for the boreal summers of years 2005–2008, the Asian mainland was representing the main source for the AMA composition ($\sim$50 %), followed by the Tibetan Plateau and The North Asian Pacific Ocean on a similar extent ($\sim$20 %).

These studies mostly rely on simulation results, therefore potentially depending on the choice of the driving model setting, while few in-situ observational evidences have been so far collected over this region (e.g. balloon soundings, Bian et al. (2012)). The StratoClim aircraft field campaign took place during the end of July - middle of August 2017 and offered a unique occasion for a detailed in-situ sampling of the air composition at the UTLS level of the AMA region during the monsoon season. In this paper, as a first instance, we exploit the observations collected during the campaign to test our Lagrangian model system. We then combine the model to the measurements to establish the effects of the convective transport on the observed UTLS air masses. Starting from the aircraft measurements, we therefore investigate the convective sources apportionment along the 8 flights, based on a Lagrangian approach coupled with geostationary observations of convective clouds. The comparison with the in-situ measurements is exploited to evaluate the capability of the back-trajectory approach to capture the convective transport, using different meteorological input (ERA-5 and ERA-Interim and kinematic and diabatic vertical velocity) and providing an evaluation of the performance of each setting. A study on the convective transport during the StratoClim campaign is then presented based on the best meteorological framework. The analysis starts from the CO species, used as a tracer for anthropogenic pollution, with the scope of characterizing the intensity and the timescales of convective transport into the UTLS. A detailed description of the transport processes and of the air composition in terms of convective tracers is presented for two selected flights for which the strongest influence of deep convection is observed, demonstrating also the potential of the system to capture the transport features in the UTLS from the scale of the anticyclonic circulation to the fine structures of pollution filaments. Finally, we present a statistical analysis based on the whole campaign flights on the vertical distribution of convective influence and age of convective air inside the AMA.

## 2 The StratoClim campaign

The StratoClim campaign took place in Nepal, with base in Kathmandu ($27°42'$ N, $85°19'$ E), between end of July and end of August 2017. Using the M-55 Geophysica aircraft, 8 flights were conducted covering the Nepalese and Northern Indian region (see Fig. 1). Flights generally lasted between 3 and 4.5 hours mainly sampling the layer between 15 and 20 km. Half of the

flights (2, 3, 5, 7) took place during local morning hours while the other half (1, 4, 6, 8) were conducted during local afternoon hours. More details on the time and altitude of the flights can be found in Figs. 7, 9 and S1–S6 of the supplementary material). Flights 1, 3, 6 and 7 (panel a in Fig. 1) probed the North Indian area and the Southern Bangladesh areas, sampling features of different origin (see Fig. 7 and S1 , S3, S6 of the supplementary material). Flights 2, 4 and 5 sampled air over the Nepalese region at different altitudes (see panel b in Fig. 1 and Figs. S2, S4, S5 of the supplementary material). Finally, flight 8 had been designed to fly over a convective system developed on the boundary between Nepal and North India (see panels c in Fig. 1 and Fig. 9).

## 3   Data and methods

### 3.1   CO and $O_3$ measurements from the Geophysica

The high time resolution (1 Hz) CO concentration values were collected by the instrument COLD2 (Carbon Oxide Laser Detector) (Viciani et al., 2018), installed in the dome on the top of the Geophysica. COLD2 is a Quantum Cascade Laser spectrometer, based on direct absorption in combination with a multipass cell. The instrument provides in-situ CO absolute concentration values with a relative error of 3 % and a sensitivity of 1-2 ppbv. The CO vertical profiles, recorded by COLD2 during the 8 flights of the campaign, are shown in Fig. 2. A background CO concentration ($CO_{base}(\mathbf{z}_0)$), in absence of fresh convective transport, is estimated taking the 5th percentile of the CO observations collected for each vertical bin of a 50 equidistant levels ($z_{0i}$) array ranging from 8 to 20 km (therefore 240 m between each bin). Ozone concentration, used for the analysis of flight 8 (see Fig. 9), was measured with a chemiluminescent ozone analyzer, FOZAN-II (Fast OZone ANalyzer) developed and manufactured by CAO (Russia) and ISAC-CNR (Italy). The instrument is described in detail by Yushkov et al. (1999). This is a fast response two-channel automated instrument to measure ozone concentration in the atmosphere from aboard the high-altitude aircraft M-55 Geophysica. This instrument makes use of solid-state chemiluminescent sensors, which are durable enough to provide continuous operation of the instrument for at least 40 hours. The instrument has a built-in high-precision ozone generator enabling periodic auto-calibration during the flight. The time response is less than 1 s and relative error $\leq 8\%$. Due to technical issues during the campaign, no $O_3$ measurements are available for flights 1 and 7.

### 3.2   Transport Analysis

In the study of atmospheric transport, the Lagrangian approach allows to characterize the source-receptor relation of atmospheric tracers. In the StratoClim framework, we used the TRACZILLA Lagrangian model (Pisso and Legras, 2008), a modified version of FLEXPART (Stohl et al., 2005; Legras et al., 2005), to understand the influence of transport and mesoscale dynamics on the features observed during the flights. Simulations have been based on the simultaneous release of a 1000 back-trajectories cluster, representative of a generic passive tracer, launched in correspondence of the aircraft position. The trajectories have been released for each second along the flight path, travelling back in time for 30 days. The trajectories are bounded to a geographical domain enclosed in a longitude and latitude range of 10 °W–160 °E and 0–50°N respectively. When

a trajectory crosses these boundaries, it is considered as terminated. The fraction of the terminated parcels corresponds to the white layer in the color-coded analysis of convective sources (Figs. 5, 7, 9, 10, 11). The back trajectories have been calculated in 4 different settings, chosing from ECMWF reanalysis horizontal winds (ERA-Interim and ERA-5 at 3h and 1h resolution respectively), and kinematic and diabatic vertical motions. Vertical diffusion is represented by a random walk equivalent to

D=0.1 $\mathrm{m}^{-1}\mathrm{s}^{-1}$ as in Pisso and Legras (2008). To individuate the encounter with convective events, diffusive back-trajectories have been coupled with high-frequency charts of cloud top altitudes from geostationary satellites (MSG1 and Himawari) as described in the following section. Therefore, a specific geographical bin is indicated as a convective source when, over that bin, a trajectory is found with a pressure higher than the high cloud top pressure, as similarly done in Tissier and Legras (2016). From this analysis we then derive an estimation of the height of convective injection, given from the altitude of encounter be-

tween the trajectories and the cloud top. In addition, the elapsed time between the convective cloud encounter and the flight measurements (corresponding to the time of trajectories release), provides an estimate of the age of the air masses. This age is representative of the time elapsed between the detrainment and the capture by the aircraft (and therefore of the age of the air in the UTLS) but we can assume it to be close the time of transport from the BL. The unknown in this analysis is indeed the time necessary for the air masses to be uplifted from the BL to the cloud top. As the sustained vertical velocity in a convective

updraft of the region is of the order of $10\ m \cdot s^{-1}$, the time span of the vertical transport from the BL can be estimated to be of few hours (1-2 hours). Finally, the possible convective sources will be classified in main source regions as shown in the region mask of Fig. 3.

## 3.3 Geostationary retrieval of cloud top: MSG1 and Himawari

The cloud top height is taken from the cloud top temperature and height (CTTH) product, developed within the European

Organisation for the Exploitation of Meteorological Satellites (EUMETSAT) Satellite Application Facility (SAF) on Support to Nowcasting and Very Short Range Forecasting (NWC) products (Schulz et al., 2009; Derrien et al., 2010). To cover the whole domain of interest, we make use of the CTTH product from both the MSG1 images, for longitudes west of 90 °E, and the HIMAWARI-8 images, east of 90 °E. The MSG1 satellite, operated by EUMETSAT and re-located at 41.5 °E after the 4th of July 2016, carries the Spinning Enhanced Visible and Infrared Imager (SEVIRI), an optical imaging radiometer. A

detailed description of MSG1 can be found in Schmetz et al. (2002). The SEVIRI instrument has 3 visible/near-infrared solar channels (0.6, 0.8 and 1.6 $\mu$m), 8 thermal infrared channels (3.9, 6.2, 7.3, 8.7, 9.7, 10.8, 12.0 and 13.4 $\mu$m) and one high-resolution visible channel (0.4–1.1 $\mu$m). The nadir spatial resolution of SEVIRI is 1 km for the high-resolution visible channel and 3 km for the others with a frequency of image collection of 15 min. HIMAWARI-8 is a geostationary meteorological satellite launched by the Japan Meteorological Agency (JMA) on the 7th of October 2014 and it is centered at 140° E, covering

the East Asian and Western Pacific regions. The visible/infrared radiometer onboard (Advanced Himawari Imager, AHI) has 16 observational bands, four bands in the visible and near-infrared spectrum (0.47 − 0.86 $\mu$m), two bands in the shortwave infrared (1.6 − 2.3 $\mu$m), one band in the medium-wave infrared (3.9 $\mu$m), and nine bands in the thermal infrared (TIR) region (5 − 14 $\mu$m). It has a nadir spatial resolution of 500 m and 1 km in the visible range and 2 km in the IR and the observations

are collected at 10 min intervals. For computational reasons, we use here one image every 20 min. For the processes we are considering, this does not affect significantly the results of the analysis.

The cloud analysis algorithm for the CTTH product includes methods for the identification and property retrieval of multi-layered cloud systems and the determination of cloud thermodynamic phase, based on an algorithm that incorporates numerical weather prediction model profiles to input vertical atmospheric profiles into a fast radiative transfer model (RTTOV from Met Office, Saunders et al. (1999)). The estimate of the cloud top height is based on different approaches, including a best fit between the simulated and the measured 10.8 $\mu$m brightness temperatures, the $H_2O$–IRW (in the IR window) intercept method (Schmetz et al., 1993) and the radiance rationing method (Menzel et al., 1983). The techniques used to retrieve the cloud top height depend on the cloud type (CT product). The CT discrimination is performed by a multi-spectral threshold method applied on the identified cloudy pixels, using the various channels combinations. This product classifies major cloud classes: fractional clouds, semitransparent clouds, high, medium and low opaque clouds. More details on the retrieval algorithm can be found in Stengel et al. (2014), Finkensieper et al. (2016) and the ATBD Meteo-France (2016) document (http://www.nwcsaf.org). A comparison fo the SAF-NWC product with spaceborne active LiDAR measurements can be found in Sèze et al. (2015). Here we use a specific version of the product where the ancillary data are taken from ERA-5 at hourly resolution. For the scope of our analysis, we selected the highest and opaque cloud classes (high opaque clouds, very high opaque clouds and very high semi-transparent thick clouds) that are representative of the deep convective events as classified in the Cloud Type (CT) product. In this study we exploited the availability of the in-situ measurements of trace gases to tune for the best use of the geostationary-retrieved cloud top. While some studies indicate generally a negative bias for the geostationary-retrieved cloud top with respect to the LiDAR-retrieved one (Sherwood et al., 2004; Hamann et al., 2014), no correction to the CTTH altitude from the SAF product appears needed for the presented analysis. A sensitivity study has been performed adding different positive biases to the cloud top altitudes from the CTTH (not shown), indicating that a correction would lead to a misplacement in the identification the convective sources. A consistent interpretation of the observed tracer enhancements is instead obtained when keeping the cloud top altitudes from the SAF without any shift. The altitude on which we are interested in this study is that at which the detrainment of convective cloud is large enough to dominate the environment and the cloud top from the geostationaries may indeed be more representative of this level rather than the optical top of the cloud as it could be seen from a LiDAR profile. The selection of the cloud types to be included in the study has also been based on the same measurements-based comparison.

### 3.4 Kinematic vs Diabatic approach and ERA-Interim – ERA-5 comparison

#### 3.4.1 Spatial distribution of sources

One of the potentially largest uncertainties in the study of Lagrangian modeling is the representation of vertical transport. The most common methods for estimating the vertical motion are the kinematic approach, that computes vertical velocities from mass conservation, and the diabatic approach, that uses diabatic heating rates as vertical velocities, in a coordinate system with potential temperature as vertical coordinate. Usually vertical velocities from reanalysis are noisy and strong dispersion was observed in the kinematic trajectories (Ploeger et al., 2010, 2011; Schoeberl and Dessler, 2011), with unrealistic transport

characteristics such as excessive or too low age of air in the stratosphere (Schoeberl and Dessler, 2011; Diallo et al., 2012). At the same time, previous studies based on ERA-Interim wind fields and heating rates in the AMA region, suggested a higher reliability of the diabatic heating rates for convective transport to the TTL (Ploeger et al., 2011; Bergman et al., 2015) as well as higher vertical motion in the inner tropical pipe region (Hoppe et al., 2016). In order to evaluate the sensitivities and estimate the uncertainties associated with vertical velocities, we compare kinematic and diabatic trajectories. Moreover, we test both ERA-Interim (horizontal resolution 1°x1°, 3 hours temporal resolution and 60 vertical levels) and the newer ERA-5 data set at higher spatial and temporal resolution (horizontal resolution 0.25° x 0.25°, 137 vertical levels and 1 hour temporal resolution). In both cases the vertical velocities are taken as instantaneous field for the kinematic computation while, for the diabatic computation, the heating rates are only available as averages on the reanalysis time step. The diabatic vertical velocities in this paper are estimated making only use of the radiative heating term, that represents the dominating contribution in the UTLS region for trajectories moving outside the clouds (Ploeger et al., 2010).

Figure 4 shows the probability distribution of all the convective sources individuated by the back-trajectories run for the whole ensemble of flights, corresponding to the different meteorological inputs (ERA-5 – ERA Interim, abbreviated respectively EA and EI) and the different vertical velocity computations (kinematic – diabatic, indicated by Z and D respectively). The comparison among the runs shows a general consistent pattern, with some differences when comparing EA with EI. In particular, EI runs show an increased probability of detecting convective sources in the maritime regions (Pacific Ocean and Bay of Bengal) with respect to EA runs, and less convective sources on the Tibetan Plateau area. A higher amount of convective sources is also found in Northern China in the EI. Comparisons between D and Z runs, for both EA and EI, does not show significant differences in the distribution of the sources. A higher percentage of detected convective parcels is found nevertheless in the diabatic respect to the kinematic computation and, while in EA the difference is of few percents units, in EI the gap between D and Z is higher ($\sim 15\%$). This difference is mainly due to the faster uplift in EID linked to the strong heating rates, while in EIZ the vertical motion is slower and more diffusive. A recent work from Li et al. (2020), shows a comparison between EAD, EAZ, EIZ, EID from two case studies, finding a faster vertical transport in EA that in EI in both D and Z runs. While this is consistent with our results in the kinematic version, we find more convective influence in the EI diabatic runs with respect to the EA diabatic. Notice however that the amount of convective influence presented here is not solely relying on the wind fields but results from the combination of vertical velocities and horizontal winds with the clouds distribution. A more systematic and detailed study on the effect of vertical transport in the reanalysis can be found in Legras and Bucci (2019).

### 3.4.2 Convective sources analysis and comparison with CO measurements

Here, we exploit the results of the StratoClim campaign to assess the performance of each reanalysis setting driving the simulations. To understand the capability of the trajectory system to reproduce small-scale transport features, we compare the temporal evolution of the simulated convective contributions with the measured CO from the COLD2 instrument. As a representative case we discuss here the performances of the trajectories analysis for F6 (6th of August 2017), while the dynamics and the features observed in this flight will be further examined in Sect. 4.1.2. During F6, the aircraft performed a long leg at a constant altitude (around 16.9 km, in the upper troposphere), traveling in South-East direction towards Bangladesh

and then turned back along the same path at the same altitude. In this transect the aircraft encountered a plume of pollution that appears as a distinguished isolated feature (panels e-f of Fig. 5, around seconds 32000 and 36000). Overall, all the four simulations indicate that the plume has been convectively injected in the upper troposphere from the South Chinese region (see panels a-d of Fig. 5). Comparing the two reanalysis results, ERA-5 shows a higher consistency in the evolution of the feature with respect to ERA-Interim. The latter seems in fact to individuate the plume at a northern position with respect to the measurements (segments 3-4 and 8-9 of the flight). Moreover, the consistency between Z and D results is not the same for EI and EA. The EID run, for instance, returns noisier results with respect to EIZ as well as a higher fraction of recirculating air (grey shade) and notable differences in the relative contribution of the sources as a function of time (panels b and d). In the case of EAD and EAZ runs instead, there are no meaningful discrepancies in the spatial and temporal structure of the plume and the main differences are found solely in the relative amount of contribution from the possible sources. According to the EAD simulations for example, the Chinese component of the plume between segments 5-6 and 7-8 represents around the 20 % of the total composition while, for the EAZ, such percentage rises to 80 % (panels a and c).

In order to have a more quantitative estimate on the quality of the different approaches, we compared the CO measurements from COLD2 with an estimated concentration derived from the simulations performed along the flight path. For each time step along the flight, the back-trajectories analysis indicates the geographical distribution of the convective sources. To take in account the spatial heterogeneity in emission intensity, we multiplied the convective distribution by the CO fluxes from an emission database. We used here the 2010 monthly emissions from the MIX v1.1 gridded emissions database (Li et al., 2017a) based on an harmonization of different up-to-date Asian regional emission inventories. The quantity we obtain ($\delta CO_{\mathrm{Trac}}(z(t))$) is therefore indicative of the CO mass potentially transported from the BL up to the flight position, hence of the enhancement of CO with respect to the background. With our diagnostic, it is not possible to pursue a rigorous computation of the CO mixing ratio since the simulation does not take in account all the processes of mycro-physics and chemistry. We therefore adopted an empirical re-scaling as explained in the following:

Figure 2 shows the total CO observations ($CO_{\mathrm{cold}}(z)$) along the vertical profiles, collected during the whole campaign. For each vertical bin $z_{0i}$ we computed the mean CO enhancement $\delta CO_{\mathrm{cold}}(z_{0i})$ with respect to the $CO_{\mathrm{base}}(z_{0i})$ baseline (as defined in Sect. 3.1).

$$< \delta CO_{\mathrm{cold}}(z_{0i}) >= \Big( CO_{\mathrm{cold}}(z(t)) - CO_{\mathrm{base}}(z(t)) \Big)_{z_{0i}<z<z_{0i+1}}$$

Similarly, for each point along the flights, we compute the $\delta CO_{\mathrm{Trac}}(z(t))$ quantity and, for each vertical bin, its average on the whole ensemble of measurements $< \delta CO_{\mathrm{Trac}}(z_{0i}) >$.

The parameters $(\delta CO_{\mathrm{Trac}}(z)/ < \delta CO_{\mathrm{Trac}}(z_{0i}) >)_{z_{0i}<z<z_{0i+1}}$ and $(\delta CO_{\mathrm{cold}}(z)/ < \delta CO_{\mathrm{cold}}(z_{0i}) >)_{z_{0i}<z<z_{0i+1}}$ are therefore representative of the relative CO enhancement, at a given altitude, with respect to the campaign average enhancement, as evaluated by the simulations and the measurements respectively.

We therefore define our CO enhancement proxy from TRACZILLA as:

$$\delta\mathrm{CO}_{\mathrm{proxy}}(z(t)) = \left( \delta\mathrm{CO}_{\mathrm{Trac}}(z(t)) \cdot \frac{<\delta\mathrm{CO}_{\mathrm{Cold}}(z_{0i})>}{<\delta\mathrm{CO}_{\mathrm{Trac}}(z_{0i})>} \right)_{z_{0i}<z<z_{0i+1}}$$

We want to emphasize that this approach is not intended to give a quantitative method for computing CO anomalies in the atmosphere. It is instead an empirical quantity that can be directly comparable to the observations to check for the correct identification of the pollution plumes.

Results for F6 are shown in Fig. 5 panels e and f. Again, the results from ERA-5 (Fig. 5 panel e) show a better consistency with the measurements and a better coherence between the kinematic and diabatic versions. The simulation presents a correct timing in the capture of the plume (between points 4 and 8), with a signal enhancement compatible with the measured one. The Pearson correlation coefficients (R) between the simulated and the measured values of CO anomalies are 65.7% and 67.7% for the kinematic and the diabatic runs, respectively. In the ERA-Interim version (panel f) the relative enhancement between outside and inside the plume is damped and the timing is not consistent with the observations. This is particularly visible in the kinematic computation (that has in fact the lowest correlation coefficient, 49.4%), while the diabatic one looks closer to the COLD2 measurements (with a correlation of 56.4%).

The results of the statistics are shown in Table ST1 of the supplement for each single flight and in Table 1 for the whole campaign average. The correlation analysis confirms that ERA-5 performs better than ERA-Interim in reproducing the trace gas transport to the UTLS ($R$ of around 60 % vs. 50%). In both EA and EI, the diabatic vertical transport reproduces better the variability observed in the CO measurements, with a slight enhancement in the correlation with data with respect to the kinematic simulations. In the EI version, though, the diabatic computation shows a higher RMSE, due to the noisy nature of the output signal. On the overall, the EAD approach appears to perform the best, with the highest $R$ correlation coefficient (60.9%), lowest root mean square error (10.6 ppbv) and lowest mean bias (3.7 ppbv) with respect to the other approaches. The interpretation of the convective transport influence that follows will therefore be based on the ERA-5 diabatic simulations.

## 4 Deep convective influence detection during the StratoClim campaign

This section provides a detailed discussion of the transport properties for two noteworthy cases of convective influence observed during the campaign: F6, which provides a clear case of deep convective injection of fresh pollution in the upper troposphere, and F8, in which the aircraft first fled over an extended continental convective system, sampling air from both fresh and old convection, and later captured an older plume of clean oceanic air injected in the UTLS by a typhoon system.

### 4.1 Flight 6, 06/08/2017: Convective outflow of Chinese pollution

#### 4.1.1 Meteorological Condition

F6 took place under the condition of unimodal anticyclone (i.e. a circulation revolving around a single centre, see panel f of Fig. S7 of the Supplementary Material). The geopotential contours at 100 hPa show a circulation centered around 33°N and

90°E, close to the flight track position (20-26 °N and 85-90 °E). F6 therefore sampled an inner part of the AMA circulation. The cold point pressure in the region of the Geophysica sampling was between 85 and 95 hPa (from ERA-5, see panel a of Fig.6), right above the level of the flight. The mean winds around the flight position were purely Easterly, transporting air from the center of South China along the anticyclonic circulation. No convective system was present close to the flight path, with the exception of the one at 92-95 °E (see panel a of Fig.6) which intensity was nevertheless not enough to inject air close to the flight level.

### 4.1.2  Air masses source apportionment

During F6 (see panel b of Fig. 7) the aircraft flew at a nearly constant altitude around 16.9 km ($\sim$ 98hPa), going from Kathmandu toward the Bay of Bengal in a South East direction. In addition, the aircraft performed a dive at 15 km over the Bangladesh coast, measuring a CO peak of 160 ppbv (at 34000 s, black line in panel c). The source contribution analysis from the trajectories (panel d) shows a dominance of North Indian air ($\sim$40 % of air composition) mixed with clean Tibetan convective air (varying between $\sim$20 % and $\sim$40 %) for the lower CO concentration regions. This air is characterized by injection at 15 and 14 km (panel e) and an average age of the order of 2 weeks and 10 days respectively (panel f). Those air masses have circulated around the anticyclone before being sampled by the aircraft. The CO mixing ratio measured at this altitude varies between 60 and 80 ppbv while flying north of 24°N but increases up to 140 ppbv when arriving around 23°N (between flight points 4-5 and 7-8, see blue line in panel c). This pollution plume is characterized by three distinct peaks. The external ones (and therefore the northern filament) reach a CO concentration of 120 ppbv (at 32000 and 36000 s) composed at 100 % by convective air of few hours of age ($\sim$ 6hours) coming from the Southeast Asia Peninsula region and the analysis reveals that this air has been injected at a very high altitude (16 km, among the highest injection altitude detected for the StratoClim campaign). Looking to the more detailed distribution of sources in correspondence of this region (panel a), we found that the convective sources over the Asian Peninsula are mainly located at the north-center of Myanmar. The analysis suggests that this filament is overlapping with another thin plume of pollution from South China, similarly very fresh (age around 1 day). The second filament is detected at around 33400 and 35500 s and brings the most polluted air (around 140 ppbv) associated by a dominance of convective South Chinese air (>80 % of contribution) with a longer average age of transport (around 2 days). This second filament originates mainly around the Sichuan basin (see the maxima around 105°E in panel a), a very polluted region of China, with an injection level of around 15.5 km. The third and weaker filament is observed at around 33000 and 35200 s and is characterized by a CO concentration of around 90 ppbv. This is composed mainly by a mixture of $\sim$17-day old Indian air (30 %) and 3-day old South Chinese air (20 %) that a more detailed analysis shows to be injected from a western region of South China (at 112 °E). The highest CO peak in the middle of the flight (flight point 6) is instead linked to very fresh pollution captured during the deep dive down to 15 km. The plume has an age of transport of the order of few hours, coming from the Pen region and South China and transported from low level injection (around 13 km). It is interesting to notice that this peak is not simply related to the change in altitude since, as shown in figure 5a, the enhancement in CO is still present even when the height-dependent background profile is subtracted.

From a more detailed analysis of the geostationary satellite images (not shown), those high injection events turn out to be very localized (convective clouds dimensions of the order of 1°) and fast developing (persisting for a period of around 2 hours) injecting air directly into the upper troposphere. This pollution is then advected horizontally by the anticyclonic circulation, at the same time ascending, following the upward large-scale pattern suggested by Vogel et al. (2019). The total vertical displacement between the convective injection of air and the moment of observation is of 2 km for the Chinese air in a time of 2–3 days and $\sim 1\mathrm{km}$ for the Southeast Asia Peninsula air in a time of 5–7 hours, faster with respect to the estimated average radiative heating rate (Wright and Fueglistaler, 2013). The analysis of this specific flight reveals how, even if during a weak phase of the convective activity, air mass samples at 16.9 km can reach up to $100\%$ of fresh convective air composition and CO anomalies up to 90 ppbv. During the whole monsoon season, deep convection may therefore play a relevant role on the composition of the UTLS, transporting young and heavily polluted air masses directly close to the tropopause level.

## 4.2 Flight 8, 10/08/2017: Tropopause crossing, overshootings and typhoon plume

### 4.2.1 Meteorological Condition

F8 took place during a phase of extended anticyclone circulation, with the core shifted westward, a mode stretching over Iran to 20°E, a central mode positioned between 60 and 80°E and a eastern mode extending to 150°E (see panel h of Fig. S7 in the supplementary material). During this flight, the aircraft sampled the inner part of the central mode, in a region where the cold point pressure was varying between 85 hPa and 90 hPa (see panel a of Fig.8). In this case, during the flight leg between the segments 3 and 8, the flight was crossing the tropopause (altitude around 86.5 – 87.5 hPa). Moreover, between the segments 2 and 5, the aircraft was passing close to the intense convective system visible in Fig. 8 panel b, south of the Nepalese borders. Winds in this part of the flight were mainly North-Easterly, due to the westward shift of the anticyclonic center.

### 4.2.2 Air masses source apportionment: convective overshoots

This flight chased the intense convective system which developed over the Gange valley in the early afternoon hours (as seen in Fig.8). The aircraft flew on a segment parallel to the Nepalese border (see panel b of Fig. 9, segments between 3 and 8) at a nearly constant altitude of 17.7 km ($\sim 86\mathrm{hPa}$) with a final segment (between points 8 and 10) at 19.1 km ($\sim 68\mathrm{hPa}$) over Nepal. It is worth noticing a progressive decrease in the CO concentration during the constant altitude segment between points 2 and 5, with values decreasing from 80 ppbv to below 60 ppbv (panel c). This corresponds, in the analysis, to a progressive decrease in the total convective influence (panel d), from a 100 % to around 50 %. On the other hand, an increase in the fraction of recirculating parcels (trajectories that travel inside the AMA region for 30 days back in time without encountering any convective influence) is detected. For this flight, the $O_3$ concentrations from the FOZAN instrument are also available (red line in panel c) showing an increasing mixing ratio from around 120 ppbv to around 150 ppbv in anticorrelation with CO. These observations suggest an increasing mixing of stratospheric air in the sampled region while traveling north. This is likely due to the vicinity of the aircraft to the tropopause, whose pressure level was close to the level of the flight, in particular right after point 4 (panel a of Fig.8), where the trajectories also find the lowest tropospheric influence. From point 2 to point 4 of the

flight, according to the trajectory analysis, the observed convective air is a mixture of 50 % of air coming from India (northern side, as visible on panel a of Fig. 9) and 25–30 % from the Tibetan Plateau. Most of the convective sources are located near the southern Himalayan barrier. This air, injected at 14 km (Tibetan air) and 15 km (Indian air) took long time to travel from the injection points to the flight position (about a couple of weeks, see panels e and f). The situation slightly changes between

points 4 and 5: the total contributions of Indian and Tibetan air decrease to less than 25 % with a contribution from the Tibetan Plateau of few percents, comparable to other minor sources, as South China. While the flight is traveling northward, it samples increasingly old air, with ages growing from 10 days to up to 20days. The recirculating air contribution in this segment is up to 20 %. This reflects in the low values of CO, that went down to 55 ppbv in this part of the flight. At the end of this segment, close to the point 5, a peak of CO is observed (reaching more than 90 ppbv). This peak is also individuated by the trajectories

that indicate a 80 % total convective activity, dominated by the contribution of the Tibetan Plateau (around 40 %). This air, contrarily to the surroundings, is injected at higher level ($> 16$km) and has a younger age (around 2.5 days). Immediately after this signature, the flight encounters an air mass of stratospheric origin, visible in the CO concentration drop to less than 50 pbbv and the corresponding enhancement in the $O_3$ measurements (up to 200 ppbv), and reflected in the decrease in the total convective influence. According to the analysis, all the air sampled during this flight was fairly old, with average ages between

10 and 20 days (with the exception of the peak close to point 5). This may seem contradictory with respect to the close position of the flight to the convective system shown in Fig. 8 panel b. The flight level, though, was quite high with respect to the main cloud top height of this system (around 86 hPa compared to cloud tops identified on average around 100 hPa). Nevertheless, a more detailed analysis reveals that, in some points, the flight was able to capture some very intense overshoots and convective outflows from very fast and localized plumes. These are hardly recorded by the infrared channels of MSG1 but their effects

are visible as small peaks in CO centered around the points 34700, 35500, and 36250 s. The comparison with the satellites visible images (not shown) suggests that those weak enhancements are associated to very young outflows, of the order of few minutes to less than one hour. A further young outflow contribution is identified around 37200 s, overlapped to the older air plume of point 5. Since the flight in this region is sampling air with an increasing background of stratospheric influence, it implies that the detected convective events are actually penetrating the tropopause level. Those events happened at a very small

temporal and spatial scale and therefore the convective analysis based on the SAF products (that have coarser resolution, see Sect. 3.3) are not always unable to discriminate them. Those very fresh outflows have important effects on the micro-physics of the UTLS and deserve a dedicated deeper analysis. A thorough discussion of these events, based on the higher resolution visible images from the geostationary satellites and additional in-situ data, will be presented in a future paper.

### 4.2.3   Air masses source apportionment: Typhoon injection

After point 5, at the same flying level as the previous segments, the measurements showed a sudden change in the trace gases concentrations. Between points 5 and 8, the flight spanned an extended region of low and nearly steady values of CO varying between 60 and 70 ppbv and $O_3$ concentration decreasing from 135 ppbv to 100 ppbv. This change is also reflected and explained by the trajectory analysis: this air appears to be associated again to a high fraction of convective influence (up to 80%) and in the specific to a dominant fraction of old ( 15 days) convective air from the South China Sea and Philippines

(SCSPhi) region ($\sim 40$ %) with sources mainly concentrated over the South Coast of China. Also seen from the geostationary BT images (not shown), those sources are related to a typhoon system named Nesat. This typhoon persisted over the ocean for several days (from the 25th to the 30th of July) and injected air at around 15 km. This system carried clean air that mixed with a fraction ($\sim 10$ %) of old ($\sim 12$ days) and likely slightly polluted air from South China and the Southeast Asia Peninsula. In this section of the flight we reach up to 80% of convective influence, the highest fraction detected at this altitude during the campaign and strongly dominated by the oceanic contribution (SCSPhi+MPac). After the point 8 the plane rose up to 19.1 km ($\sim 68$ hPa) entering into the stratosphere. This reflects in a sharp increase in the $O_3$ concentration (up to 400 ppbv) and decrease in the CO mixing ratio (between 20 and 30 ppbv). No convective influence is found in this flight segment. According to the trajectory analysis along this ascending segment and the following descent, the influence of air from the typhoon injection extended from 17 km up to 19 km. While, during the whole campaign, the other convective contributions at this level were on average 20 days old, this case represents an exception with a large fraction of air masses related to an age of 11-12 days.

### 4.3 Average convective influence for the whole campaign

A similar analysis was carried out on all the campaign flights. Figure 10 shows an overview of the main observed convective sources. The campaign took place in a break phase of the monsoon, characterized by less precipitation and less convective activity over the Indian sub-continent with respect to the average, except for the last days (after the 6th of August) when some intense (but isolated) convective systems were observed. This is also reflected in the total convective influence observed by the different flights. In the flights F2 through F5, when a large part of the track was above 17 km, the observed convective influence was limited to less than 50% and to around 60% for F5, with a strong dominance of local air, coming from the Indian subcontinent, especially the northern part, and the Tibetan Plateau (also visible in panel c of Fig. 4). In the first and the last three flights a larger variability of sources was instead observed, with a total convective contribution larger than 70%. F6 and F7 also show a non-negligible contribution from the South-China region (dominated by convection from the Sichuan area and the central-western China, panel c of Fig. 4), bringing polluted air to the UTLS level. Contrarily to the expectation, almost no maritime air was observed, except for the last flight F8, that sampled the outflow from the typhoon mentioned in Sect. 4.2.2 (dark orange shade in Fig 10). On the overall, during the campaign the Geophysica sampled a prevalence of convective air from the Indian subcontinent with an average influence on the air composition of around 20-30 %. Another big contributor is represented by the Tibetan Plateau. The observed convective influence is strongly dependent on the region investigated by the aircraft. We observed in fact a higher contribution of the Tibetan Plateau (up to 30 %) during the flights that were operating over Nepal. Two additional relevant sources of convective air are South-China and Pakistan, with a contribution varying according to the synoptic conditions and the sampled region. In correspondence of both source regions we generally observed an enhancement in pollutants concentration, both being representative of highly emissive areas. The total contribution from regions other than those selected in the mask of Fig. 3 is, for each flight, negligible (black shade, between 0.2 % and 0.8 %). Panel a of Fig.11 illustrates the convective influence along the vertical, as observed by the totality of flights, compared with the correspondent observed mean CO (panel b) over the same level bin. The analysis shows that, on average, below 15 km the sampled air is mostly of convective origin (close to 100%) and very young (age below 3 days). It should be noticed, however,

that the total measurements at those heights are below 1000 samplings (that means less than 1000 seconds of measurements out of around 93000 seconds of total data collection), therefore the relative source apportionment may be not statistically significant with respect to the above levels (see the vertical distribution of sampling in Fig. S8 of the Supplementary Material). Between 15 and 17 km, close to the level of the tropopause, convection still plays a predominant role with a percentage of
influence greater than 90% and an average time of transport of the order of one week. In this layer, the average CO anomaly is of about 25 ppbv over the background (that we again estimate by the 5 percentile, dark grey shade), with higher values reaching 60 ppbv. Those peaks are mainly related to high altitude convective injection from polluted regions like Pakistan and South China, that represent a relative contribution between 10 to 20 % of the air composition. Above 17 km the convective influence rapidly decreases with height, passing from 90% at 17 km, with average age of 11 days, to 45 % at 18 km, with ages of around
20 days. The mean CO anomaly here is of around 15 ppbv with maxima up to 30 ppbv. At 19 km, the convective influence is almost negligible and the CO enhancement decreases to mean values of 10 ppbv, with maxima of around 15 ppbv.

## 5   Discussion and conclusions

The StratoClim campaign offered, for the first time, a comprehensive dataset of in-situ measurement in the UTLS, inside the summer AMA. It represented a unique occasion to investigate the details of deep convective transport into the low stratosphere
and a useful reference dataset to evaluate transport models performance over this region. We therefore tested the quality of the TRACZILLA dispersion model fed with different reanalysis (ERA-5 vs. ERA-Interim) and vertical motion settings (diabatic vs. kinematic), against the measurements of the CO species. We found that ERA-Interim, with respect to ERA-5, generally recognized more sources over the maritime regions (i.e. Pacific Ocean, Bay of Bengals) and Northern China and less from Tibetan Plateau. The comparison with the CO measurements shows that the transport in the AMA region is better
represented by the ERA-5 winds. The diabatic runs, in both reanalysis, are indicating a higher percentage of contributing convective events, with larger differences in the ERA-Interim setting. This is consistent with previous results that, for the upwelling regions of the inner tropics, indicates higher vertical velocities in the diabatic computations (Hoppe et al., 2016; Garny and Randel, 2016; Ploeger et al., 2012). The comparison with COLD2 CO measurements indicates a better correlation with the observed variability in the diabatic with respect to the kinematic vertical motion for both ERA-5 and ERA-Interim.
While in ERA-5 all the correlation parameters improve in the diabatic version, in ERA-Interim the diabatic ascent is associated to the highest root mean square error. Overall, the ERA-5 diabatic version demonstrates to be the closest representation of transport for the UTLS level in the AMA region. In general, the trajectories-satellites system proves to be able to describe the convective events consistently with the observations, well capturing both the general circulation and the small and short-lasting CO transport features. Using this setting we therefore investigated the details of two flights of the campaign in which we
observed extended plumes of deep convective outflows. In the first case, the 6th of August 2017, a convective plume carrying high CO concentration (up to 140 ppbv, estimated to be around 70 ppbv over the background concentration) was observed at 16.9 km ($\sim$ 98 hPa). The source analysis indicates that the plume has an almost exclusive convective origin, being a mixture of a very fresh plume (order of few hours) of polluted air coming from the north of Myanmar, directly injected close to the

level of the flight, and a more extended plume of highly polluted air from the Sichuan Basin that entered the UTLS around 2 days before. Both Myanmar and the Sichuan Basin are indeed among the highest densely populated and polluted regions of South East Asia (Aung et al., 2017; Ning et al., 2018; Qiao et al., 2019) well known for being characterized by heavy convective precipitations (Romatschke and Houze, 2010; Liu et al., 2018; Li et al., 2017b). The second case, the 10th of

August, captured the outflow of a large convective system over the Gange Valley, another region characterized by convective precipitation maxima (Kumar, 2017). While the flight was indeed traveling above the cloud tops, the altitude of sampling (17.7 km $\sim$ 86 hPa) was too high to capture the mean fresh outflow in its whole. Most of the air was of convective origin (between 50 to 100 %) but on average related to recirculating air injected at around 15 km and slowly uplifting (order of 10 to 15 days) to the level of the flight. Nevertheless, a clear signature of high CO mixing ratio from fresh injection has also been

identified, over-imposed to a general decreasing trend due to a gradual entering into the stratospheric regime. Those peaks are due to individual intense towers of convection, part of the extended convective system, that injected very fresh air close to the level of the flight (17.7 km). While one of those towers is captured by the trajectory analysis, the spatial extent of the others is too small (order of few km) to be clearly identified by our simulations based on the satellite IR images. Those events are instead detected by the higher resolution images of the visible channel of the geostationary satellites and deserve a separate

study. On the overall, the campaign was successful in capturing episodes of convective outflow and deep convective overshoots over the AMA region, even if in a break phase of the monsoon. The main sources of the sampled air was traced back from Northern India and the Tibetan Plateau, in line with previous model studies. Those sources though, in the framework of the campaign, are not usually linked to high enhancement of CO and are mostly related to old recirculating air (order of couple of weeks) due to the campaign main area, located upwind to the main pollution sources in North India, in the easterly branch

of the anticyclone. Higher CO concentration was instead detected in correspondence of young ($\sim 1 - 2$ days) air from South China and the Southeast Asia Peninsula. The convective events during the period of the campaign appear to reach the cold point pressure level frequently, with times of transport in the order of one week on the average and few events of direct quick injection. Above the tropopause level ($> 17$ km), the analysis reveals a still significant convective influence, with average time of transport of $\sim 20$ days and bringing CO anomalies of the order of 15-30 ppbv. These values are comparable to the mean

anomalies estimated in previous model studies at the same level (e.g. Pan et al. (2016); Barret et al. (2016)). It is important to point out that the convective sources influences observed during the campaign are strongly related to the position and time period of the campaign itself. The region spanned by the aircraft is limited to the central-southern part of the anticyclone and the sources observed are therefore mainly the ones that are found upwind of the anticyclonic circulation with respect to the aircraft position. Similarly, a sampling region located more South would have likely captured more maritime convection

(that is expected to be dominant with respect to the continental contribution for extension and frequency) and an additional prolongation of the campaign period after the break phase would have probably allowed to sample more intense convective events. This analysis nevertheless provided an in-situ measurement assessment of the combined satellite-modeling approach to represent the convective transport in the region, providing a tool for a reliable analysis on a longer period and a wider region. On the other hand, the analysis of the StratoClim flights provided evidence on how the convective events over these regions, even

if very short-lasting and localized, may be intense enough to allow a fast and direct injection of highly polluted air at the UTLS

level, spanning a large domain, that can then keep rising to enter the stratospheric circulation. As the campaign was conducted in a phase of weak convective activity and having spanned only a limited region of the AMA, it is reasonable to expect the occurrence of more intense and frequent events of fresh and eventually polluted air into the UTLS along the whole season. Those events may in principle strongly impact the chemical composition of the stratosphere. Some of these intense convective

injections, while bringing polluted air at very high levels (peaks detected at 17.7 km up to 50 ppbv over the background) play also a role in the hydration-dehydration of the stratosphere. The discussion of the hydration effects, based on the analysis of the more resolved visible images, and the analysis on the frequency and seasonal relative impact of deep convection on wider time/space scales, are matter of ongoing investigation.

*Data availability.* Data will be freely available at the https://halo-db.pa.op.dlr.de/mission/101 database from the end of June 2020, in the
meanwhile they will be available upon request to the authors

*Author contributions.* S. Bucci performed the analysis and wrote the draft. B. Legras provided the trajectories simulations and the tool for the convective analysis. F. D'Amato, S. Viciani, A. Montori, A. Chiarugi provided the carbon monoxide measurements, F. Ravegnani and A. Ulanovsky provided the ozone data. P. Sellitto, F. Cairo and F. Stroh participated in the redaction of the paper and provided useful comments and insights.

*Competing interests.* The authors declare that they have no conflict of interest.

*Acknowledgements.* This work was supported by the StratoClim project by the European Community's Seventh Framework Programme (FP7/2007–2013) under grant agreement no. 603557, CEFIPRA5607-1 and the TTL-Xing ANR-17-CE01-0015 projects. We thank the whole StratoClim Team for having made this successful campaign possible. Meteorological analysis data are provided by the European Centre for Medium-Range Weather Forecasts. ERA-5 trajectory computations are generated using Copernicus Climate Change Service Information.
We also thank the AERIS data and service centre for providing access to the MSG1 and Himawari data.

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

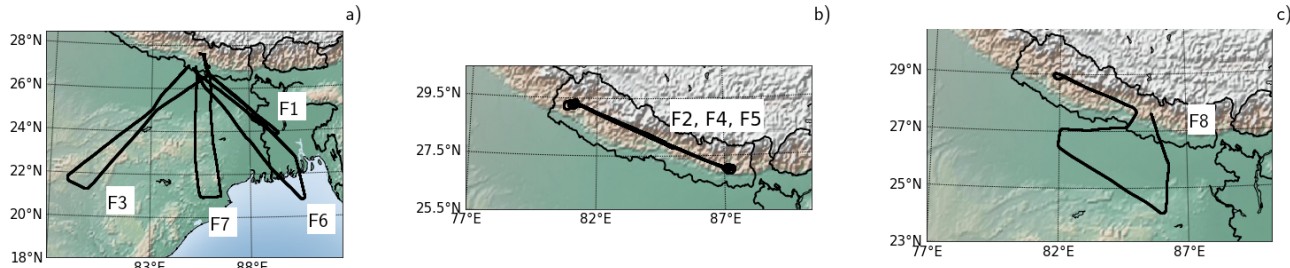

**Figure 1.** Campaign flight tracks. Panel a): flights 1 (29th of July 2017), 3 (31st of July 2017), 6 (6th of August 2017) and 7 (8th of August 2017). Panel b): flights 2 (29th of July 2017), 4 (2nd of August 2017) and 5 (4th of August 2017). Panel c): flight 8 (10th of August 2017)

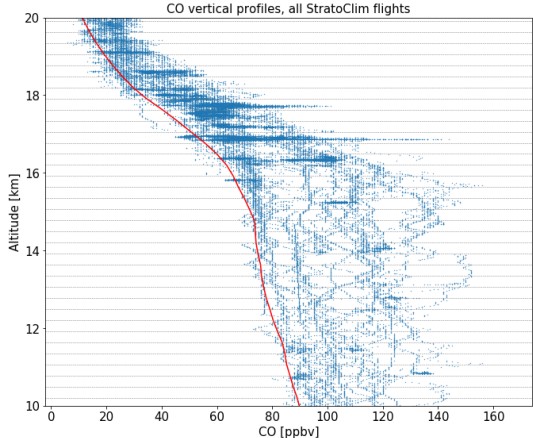

**Figure 2.** Total CO measurements along the whole 8 flights of the campaign. The red line represents the estimated CO background, computed as the 5 percentile of the measurements along the 240 m altitude bins (horizontal gray thin lines)

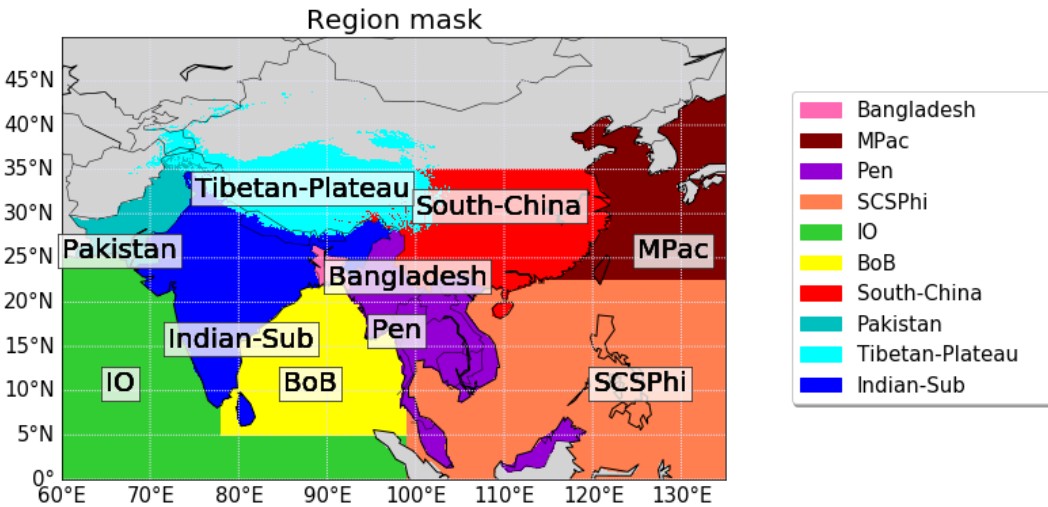

**Figure 3.** Source regions mask for the AMA area. In color: Indian Subcontinent (Indian-Sub), Tibetan Plateau (Tibetan-Plateau), South China (South-China), Pakistan (Pakistan), Southest Asia Peninsula (Pen), South China Sea and Philippines (SCSPhi), Bangladesh (Bangladesh), North China (North-China), Bay of Bengal (BoB), Mid Pacific (MPac), Indian Ocean (IO), Other sources

| | | All Flights | |
| --- | --- | --- | --- |
| | Correlation R [%] | RMSE [ppbv] | Mean Bias [ppbv] |
| EIZ | 51.2 | 13.0 | 4.3 |
| EID | 52.6 | 16.4 | 4.2 |
| EAZ | 58.8 | 11.0 | 3.7 |
| EAD | 60.9 | 10.6 | 3.7 |

**Table 1.** Correlation coefficient, Root Mean Square Error and mean bias between $\delta\text{CO}_{\text{proxy}}(z(t))$ and $\delta\text{CO}_{\text{cold}}$ for the different meteorological settings (EAZ,EAD,EIZ,EID). Values are averaged on the results from all the 8 flights.

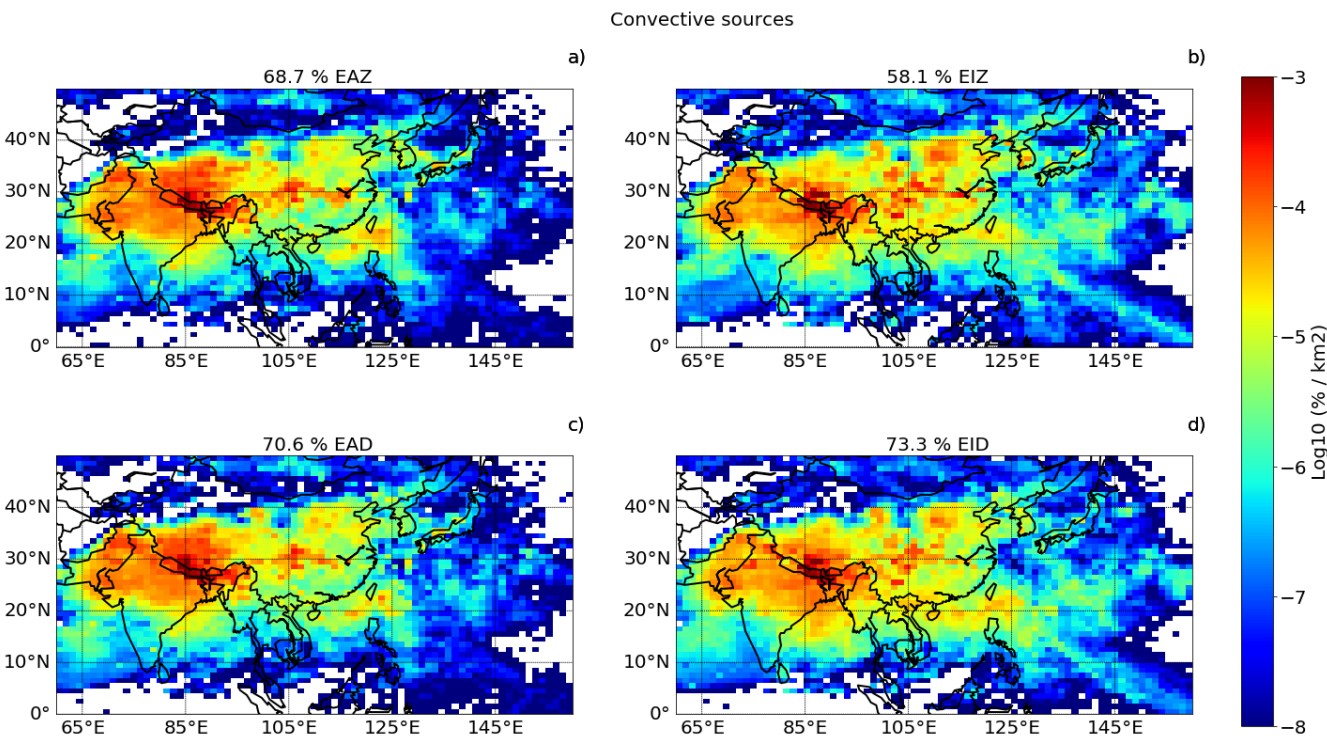

**Figure 4.** Convective sources distribution identified by the trajectories for the whole aircraft campaign. The probability of finding a convective source in a specific $km^2$ is computed as the fraction of parcels that encounters there a convective cloud, with respect to the total number of released parcels. The plots are in logarithmic scale. The percentage in the title indicates the fraction of convective parcels integrated over the whole surface. Panel a) Convective sources distribution obtained with ERA-5 meteorology and kinematic vertical motion (EAZ); panel b) ERA-Interim meteorology and kinematic vertical motion (EIZ); panel c) ERA-5 meteorology and diabatic vertical motion (EAD); panel d) ERA-Interim meteorology and diabatic vertical motion (EID)

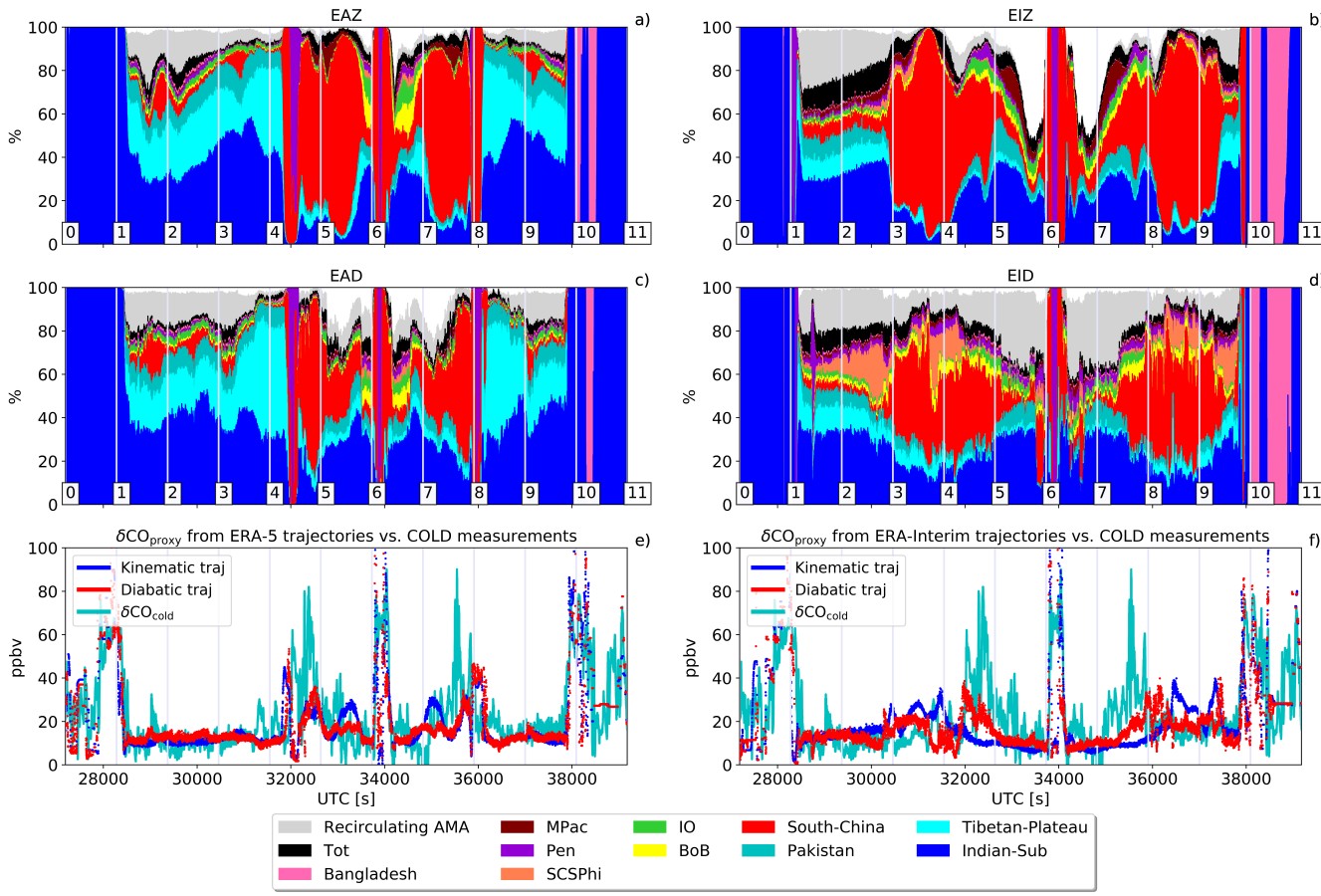

**Figure 5.** Convective sources contributions in the ERA-5 (left column) and ERA-Interim (right column) computations along the flight time of F6 (6th of August 2017). The thickness of each colored layer in panels a-d represents the percentage of contribution of the region associated to the color code showed in Fig. 3. The grey layer indicates the percentage of parcels recirculating inside the Asian Monsoon Anticyclone without hitting convection. The thickness of the black layer indicates the percentage of contribution from sources different from the ones identified in Fig. 3. The remaining white layer represents the percentage of parcels that exited the boundaries of the domain before encountering any convective cloud. Panel e) and f) shows the time series of the CO anomalies $\delta CO_{cold}$ (in green) measured by the COLD2 instruments compared to the artificial CO enhancement $\delta CO_{proxy}$ simulated from both the kinematic (blue) and diabatic (red) computations. Numbers on the panel punctuate the flight path on equally spaced time intervals, corresponding to the position of the flight, like shown in Fig. 7 panel b

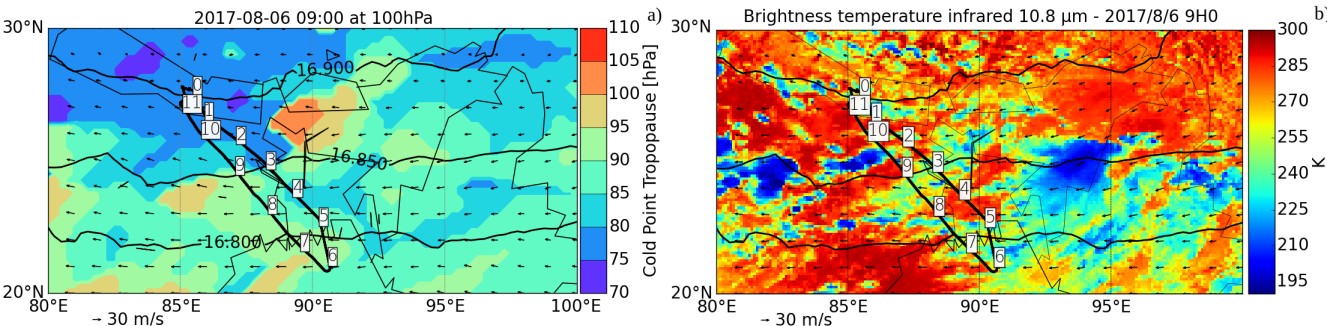

**Figure 6.** Panel a): Cold Point Pressure, wind speed and direction and geopotential contours from ERA-5 at 100 hPa for the 6th of August at 09:00 (around the middle of flying time of F6). Panel b): Brightness Temperature at 10.8 $\mu g$ from the MSG1-Himawari observations at the same time as in panel a). The black line traces the complete flight track. The numbers on the track punctuate the flight path on equally spaced time intervals, corresponding to the numbers shown in Fig. 7 panel d

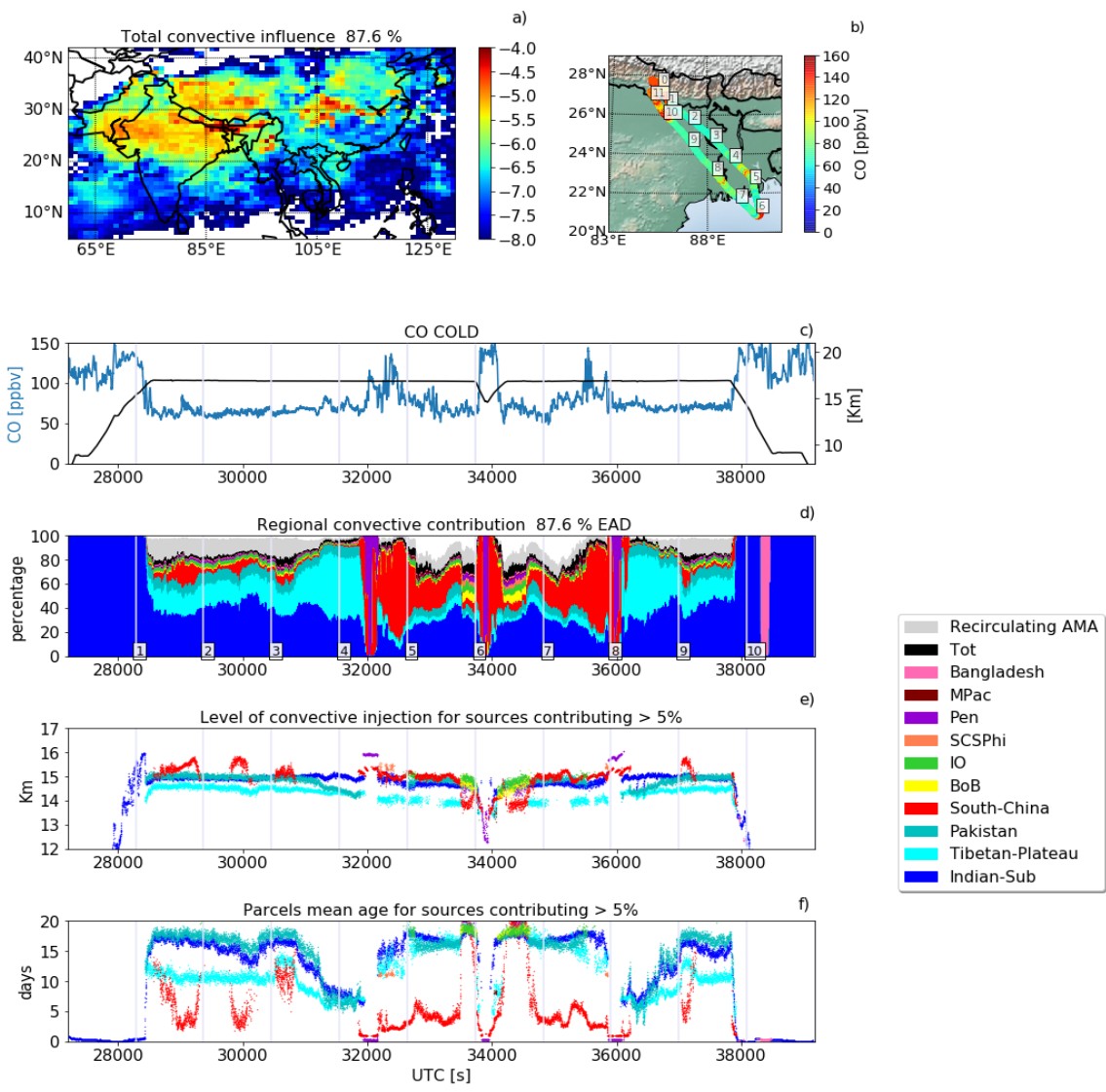

**Figure 7.** Back-trajectories analysis of convective sources for F6 (6th of August 2017). Panel a) Convective source regions distribution. Panel b) CO concentration along the flight track. The numbers along the flight track correspond to the numbers along the time series of panel d. Panel c) CO concentration along the path of flight (in blue) and altitude of the flight (black). Panel d) Convective sources contributions along the flight. The color is referring to the region color code of Fig. 3 plus the non-convective air recirculating inside the AMA (grey shade) and the remaining convective sources (black). Panel e) Level of injection for convective sources contributing for $5\%$ of the total convective air. This level is computed as the height at which the trajectory is first found below the convective cloud top. Panel f) age of the convective air for convective sources contributing for $5\%$ of the total convective air. The age is computed as the number of days between the trajectory release and the encountering of a convective cloud.

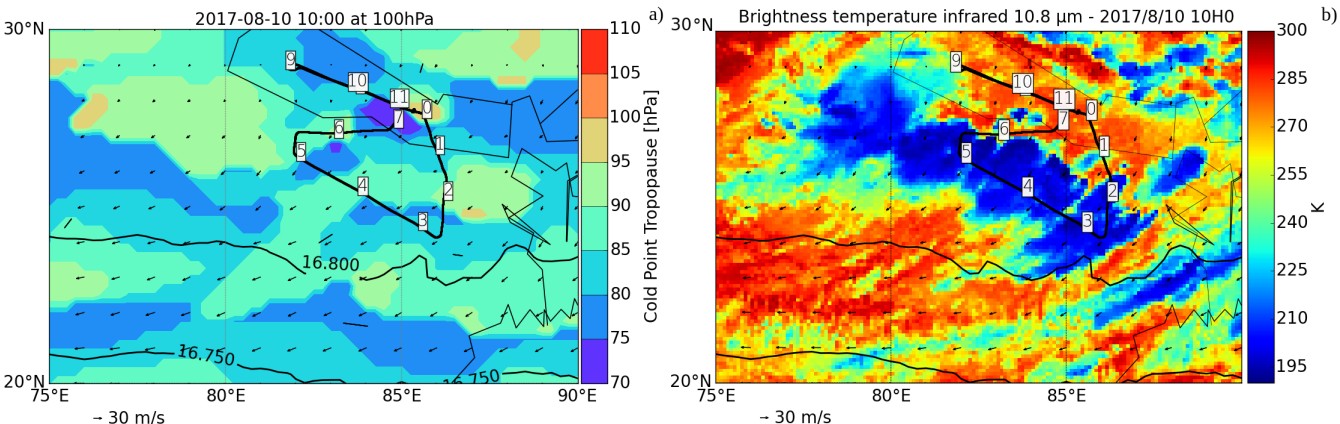

**Figure 8.** As in Fig. 6 but for F8 (10th of August 2017) at 10:00 (around the middle of the flight).

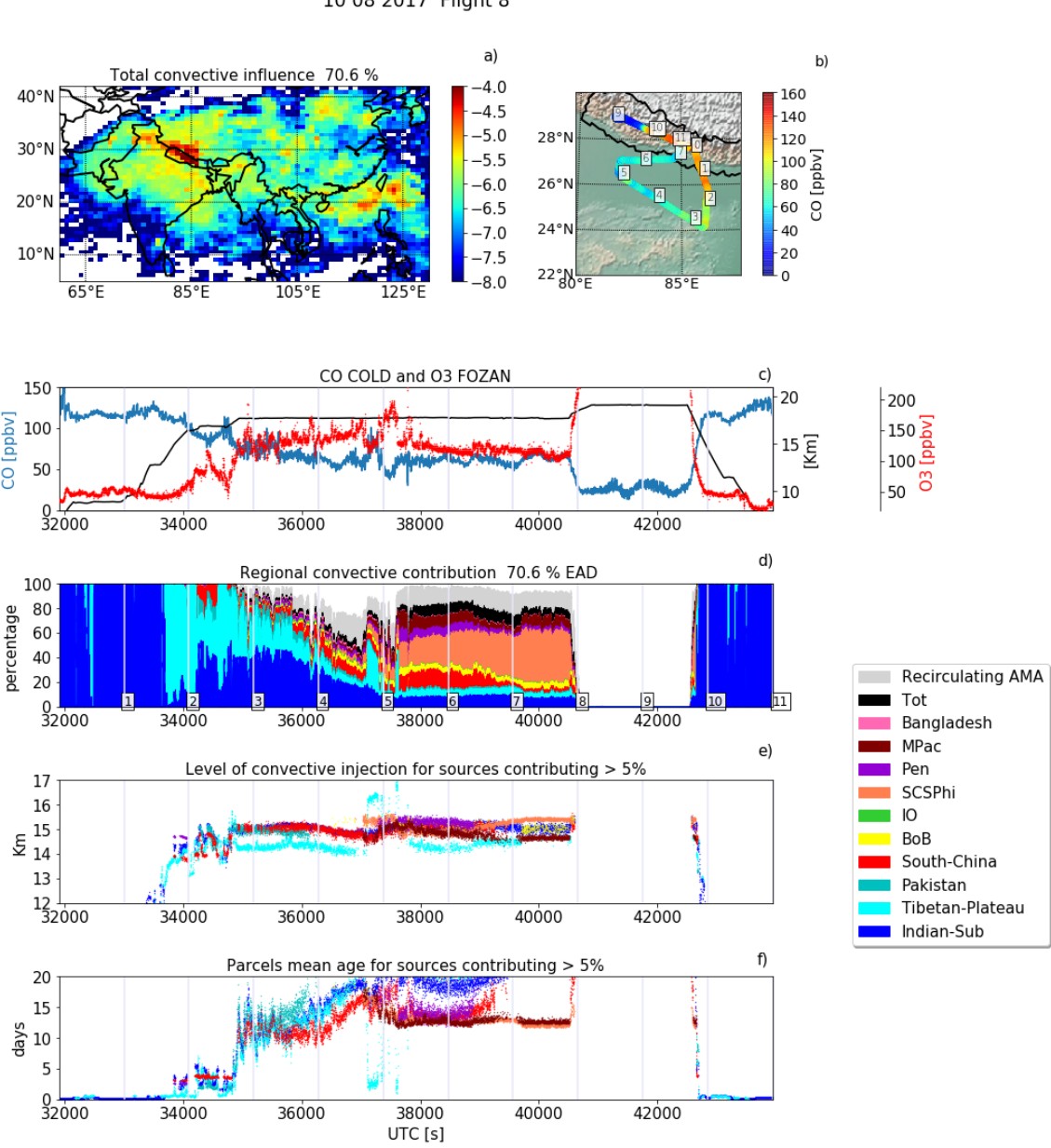

**Figure 9.** As in Fig. 7 but for F8 (10th of August 2017). Panel c) reports also the $O_3$ concentration from the FOZAN instrument (not available in F6)

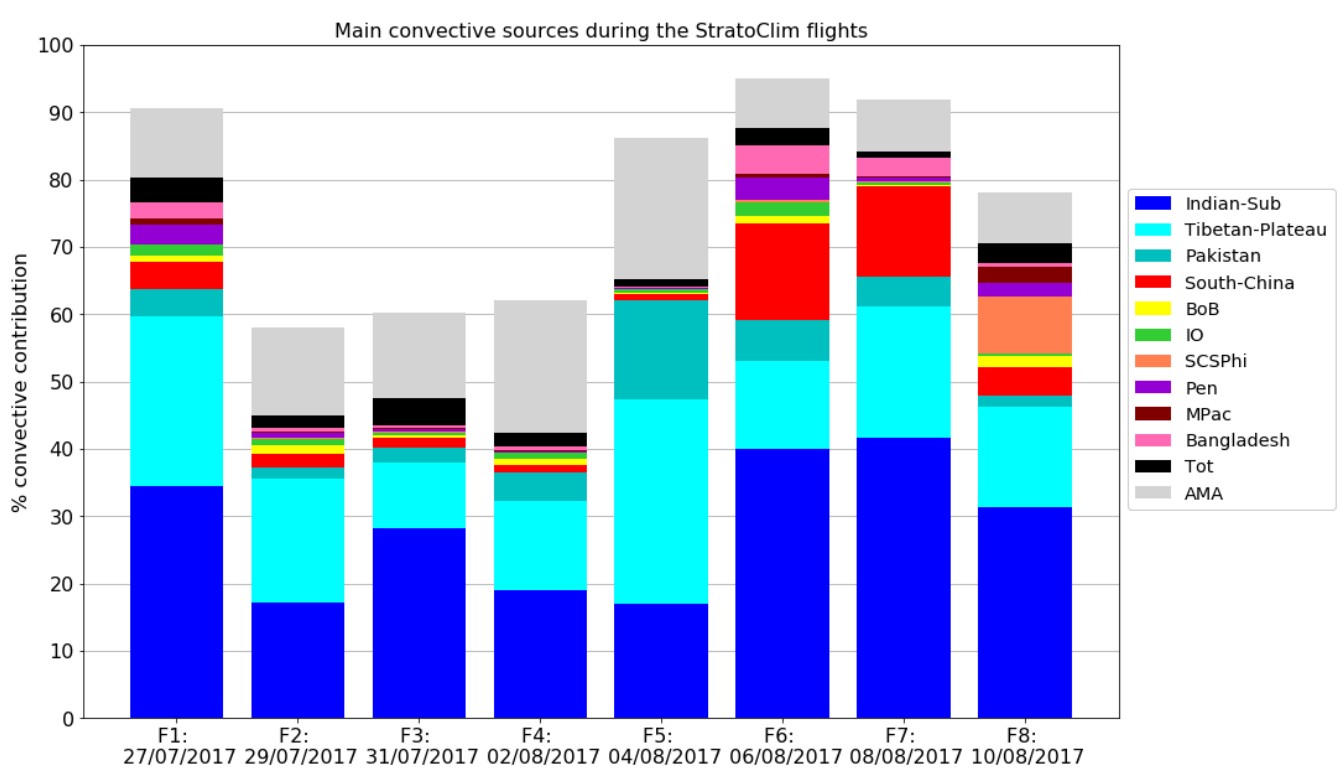

**Figure 10.** Overall convective contribution, and source regions, for each single flight. Colors refers to the mask of Fig. 3

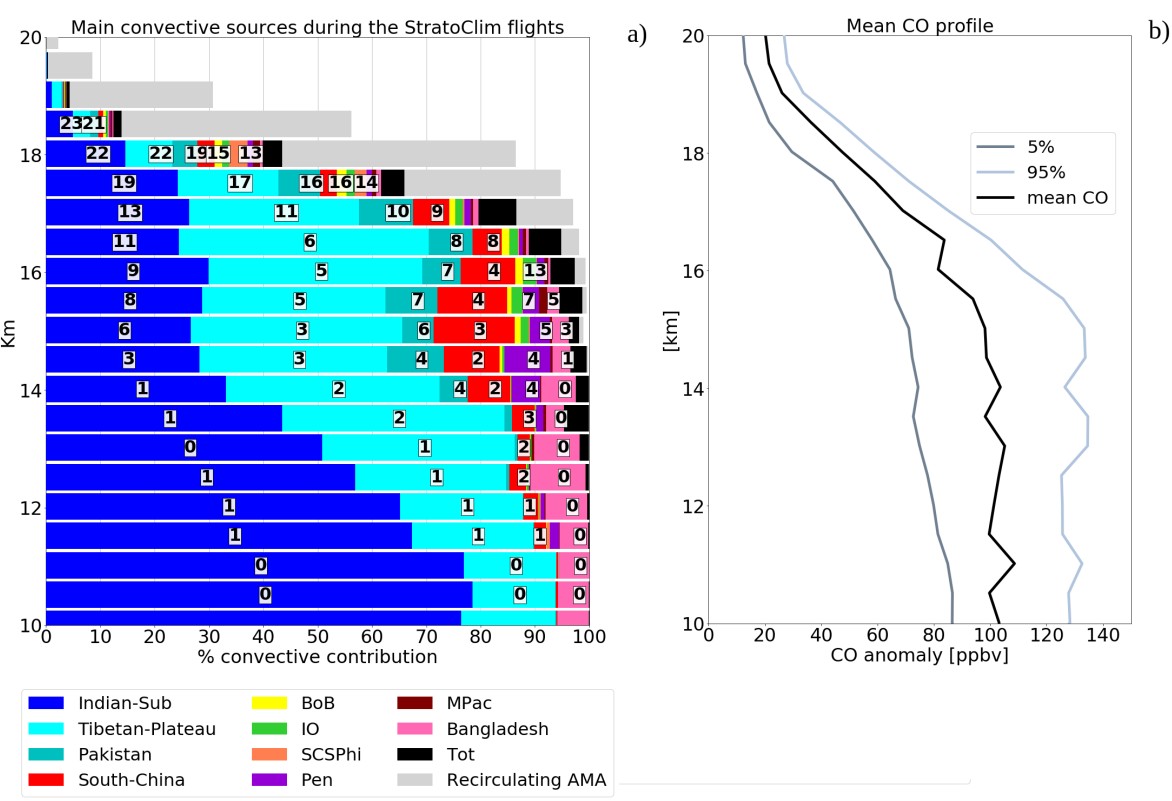

**Figure 11.** Panel a): vertical distribution of convective influence as individuated from the trajectories through the whole campaign. Colors indicate the different regions as in Fig. 3 and the number indicate the mean age of transport for each source region at that height bin. Panel b): mean CO concentration (black), 5 (dark grey) and 95 percentiles (light grey) for the different height bins, as seen by the COLD2 instrument through the whole campaign