# Peer review of "Deep convective influence on the UTLS composition in the Asian Monsoon Anticyclone region: 2017 StratoClim campaign results"

_Atmospheric Chemistry and Physics, 2019_

## Referee Comment (RC1) · Anonymous Referee #1 · 5 Feb 2020

Based on the StratoClim aircraft observation and back-trajectory modelling, this manuscript investigated the convective injection of trace gases into the UTLS within the Asian monsoon anticyclone. The convective sources in the air parcel samples were diagnosed by integrating the trajectories output with high-resolution observations of cloud tops from geostationary satellites. The larger influence by convective injections was found from the continental sources of China and India for all the 8 flights during the campaign over Nepalese and Northern Indian region. Observed thin filamentary structures of higher CO air were mostly associated with young convective air and a predominant South-China origin. These results will enhance understanding the transport of surface pollutants to the stratosphere during the Asian monsoon region. I

recommend its publication in ACP before two major issues are resolved.

Major issues:

1. In this manuscript, the convective source is identified when the trajectory is found with a pressure higher than the high cloud top pressure, and the highest and opaque cloud classes are representative of the deep convective events. The features of cloud top height will determine how the convective contribution changes with height, as discussed in the later part of this manuscript. So whether the cloud top heights of these highest and opaque cloud classes can represent the deep convective main outflow levels is a critical issue. I'm wondering the uncertainty in this issue. The manuscript should have some detailed description about the uncertainty.

2. The convective sources diagnosed in this manuscript are based on the StratoClim flight tracks, which only cover a small part of the Asian monsoon anticyclone, and to the south of the anticyclone center. Whether the convective sources diagnosed here are different from other regions and can represent the whole anticyclone is a critical issue, which should be considered with caution. The manuscript should discuss this issue.

Minor issues:

1. Page 2 Line 7: This statement is valid in the upper troposphere and lower stratosphere.

2. Page 2-3: These papers (Chen et al., 2012; Bergman et al., 2013; Vogel et al., 2015) have different targeted parcels (within the AMA or not) and therefore different sources, which should be noted.

3. Page 3 Line 8: In situ balloon soundings over the Tibet Plateau by Bian et al. (GRL, 2012) could be mentioned here.

4. Page 6 Lines 19-30: The comparison among EAD, EAZ, EID and EIZ is also done by Li et al. (ACPD, 2019), which could be cited here.

5. Page 7 Line 5-6: The plume is located at #6 at 34000s UTC?

6. Page 9 Line 5: F6 therefore sampled the inner part of the AMA. Whether the parcels sampled during F6 are called "inner part" depend how the inner part of the AMA is defined. The 100 hPa circulation is centered around 33N, while the flight track position is 20-26 N, where is 7-13 degrees from the center. China sources make a critical contribution to the south part of AMA, which has also been shown by Yan & Bian (AAS, 2015).

7. Page 9 Line 6: The cold point pressure is derived from ERA5?

8. Page 9 Line 7-8: flight. The mean winds around the flight position were purely Easterlies, transporting air from the center of South China along the anticyclonic circulation, which is also the main reason why South China contributes a lot in the convective sources as discussed in Page 7. This issue should be mentioned there.

9. Page 9 Line 11-12: I'm curious if there were no aircraft dive here, would the CO peak be observed?

Refrences:

Bian, J. C., Pan, L. L., Paulik, L., Vömel, H., and Chen, H. B.: In situ water vapor and ozone measurements in Lhasa and Kunming during the Asian summer monsoon, Geophys. Res. Lett., 39, L19808, doi:10.1029/2012GL052996, 2012.

Li, D., Vogel, B., Müller, R., Bian, J., Günther, G., Plöger, F., Li, Q., Zhang, J., Bai, Z., Vömel, H., and Riese, M.: Dehydration and low ozone in the tropopause layer over the Asian monsoon caused by tropical cyclones: Lagrangian transport calculations using ERA-Interim and ERA5 reanalysis data, Atmos. Chem. Phys. Discuss., https://doi.org/10.5194/acp-2019-816, in review, 2019

Yan, R. and J. Bian, 2015: Tracing the boundary layer sources of carbon monoxide in the Asian summer monsoon anticyclone using WRF–Chem, Adv. Atmos. Sci., 32, 943–951, doi:10.1007/s00376-014-4130-3, 2015.

---

## Referee Comment (RC2) · Anonymous Referee #2 · 2 Mar 2020

Summary: This work undertakes a comprehensive analysis of the influence of convective transport on the UTLS airmass in the Asian monsoon region using trajectory modeling and in situ measurements during a Southern Asia aircraft campaign. The authors examine the effectiveness of kinematic and diabatic vertical velocity in two different reanalysis by comparing reconstructed CO from a trajectory model to in-situ aircraft observations of CO. The results show that the ERA5 product produces a more realistic representation of the observed CO compared to the ERA-Interim product. The diabatic vertical velocity option is also found to perform slightly better compared to kinematic vertical velocity. The authors break Asia into several regions to investigate the regional origin of air masses sampled during the aircraft campaign with the aid of

satellite observations of convective clouds. Two flights during the mission are examined in detail to determine the convective origin of the air masses as well as their age and altitude of cloud interaction. Statistics of convective origin for all eight flights of the campaign are also presented, organized by both flight day and measurement height.

General comments: Although the manuscript reports significant research results that are important for interpreting the StratoClim campaign data, and has the potential of becoming part of the work contributing to new scientific insights on the role of Asian Summer Monsoon in atmospheric chemistry and climate relevant processes, the current version has significant deficiencies. Scientifically, the manuscript needs to be revised to relate the study to outstanding scientific questions and to speak to scientific researchers beyond the StratoClim team. Presentation wise, the manuscript needs to be revised to have a structure that allows the readers to get the take home messages. For these reasons, we do not consider the current version of the manuscript as meeting the standard of publication quality. The revisions we recommend are detailed below.

Major comments:

1) Scientifically, the manuscript does not adequately define the objectives of the study in the scope of the outstanding scientific questions. As a result, it is not clear what the key conclusions are. The only clear conclusion is related to the better performance of ERA-5 and diabatic vertical wind as opposed to ERA-Interim and kinematic wind. This conclusion does not support the title of the article.

The lack of a clear objective is also reflected in the description of the depth of convective "injection" and the "age" of air in the samples. First of all, no clear definition is given to define "age". We assume it is defined by the length of the back-trajectory from flight track to convective top encounter, but what is the physical significance of this quantity?

To resolve these issues, we have the following specific suggestions:

a. Revise the introduction to clearly state the objectives. If identifying the chemi-

cal characteristics of convectively transported air from different regions is the ultimate goal of the data analysis, but the composition analysis is beyond this work alone, it still needs to clearly articulated. The regions used for "airmass origin apportionment" should also be defined with different chemical emission characteristics or convection behaviors in mind.

b. An implicit goal of the work is to investigate how often and how deep the convective transport is influencing the "UTLS" composition. Separating the UT from the LS is important. If the tropopause identification is not supported by the flights themselves, an estimate using ERA5 data could still be very helpful to quantify and characterize the height of convective "injection" relative to the tropopause. With the help of the tropopause location, quantifying the direct influence of convection relative to the tropopause, the relative contribution from the regions in the UT, and when and where convection influenced the stratosphere can be a significant conclusion of this work.

2) From the presentation point of view, the manuscript suffers from a deficiency of too many details and lack of clear take home message. Although there are many details highlighting convective influence from different boundary layers (such as Northern India and the Tibetan plateau), there is no clear message why transport from these regions is important. For the two selected flights described segment by segment, the writing style is similar to that of a detailed flight report. Convective origin, or a sample's "airmass source apportionment", is a big focus of the analysis, but no significant chemical consequences are shown from the analysis or articulated in the introduction. After all these details, it is not clear what the significant findings are or what is scientifically new. We suggest that the discussion and analysis be reorganized around new findings.

3) A number of sections are written as one paragraph. It seems largely due to the style of "flight logging" used throughout. This poses a challenge for the readers. We suggest that the authors highlight the main goal of each discussion, select significant details, and break the sections into a number of paragraphs according to the take-home messages.

4) We also suggest that the authors improve the figures. Specifically:

a. All the key information would need to be in the paper, not the supplement. For example, it is important to show the flight tracks relative to the flow pattern of the anticyclone.

b. It is also a good practice to make the figures, including the titles and axis labels, large enough to read in the printed version. There are a number of issues with this, including Figures 3, 6 and 8.

c. The large number of regions defined in Figure 3 should be re-considered since the authors seem to have run out of colors to represent all the regions distinctly. For example, in the later figures, Tibetan plateau, MPac, Bangladesh and Pakistan are not always separable, especially in the print version.

Specific comments:

1) We suggest that the authors address the issue of CO being the only chemical convectively influenced composition variable shown in the manuscript, since the paper is about the convective influence on the UTLS composition. We note that the use of reconstructed CO to diagnose the trajectory based convective transport identification is a nice piece of analysis in this work. CO alone, however, doesn't represent the objective of the campaign. It would be good to state the limited objective of using CO in this analysis and the the goal of the analysis is to support the full scope of chemical composition analysis, in particular the short lived active species, etc.

2) A statement about the magnitude of uncertainty in satellite-derived convective cloud tops would be beneficial since the results hinge very strongly on these being accurate.

3) If the assumption is made that once a parcel encounters a convective cloud top it is considered to simultaneously contact the boundary layer, it is an important assumption to be explicitly stated and justified.

4) P11 L12-14: It is an inaccurate statement of "stratospheric intrusion" based on the

observed CO-O3 structure without a tropopause analysis. It is also possible the flight sampled a filament of stratospheric air produced by the large scale stirring.

5) P4 L23-25: "The trajectories move . . ." needs to be revised. This sentence has no clear meaning. Do the authors intend to say "Only the trajectories moving within the domain 10-160 E and 0-50N are considered" in the analysis?

Technical comments:

There is an inconsistency between Sections 1 and 2 about when StratoClim ended (beginning or middle of August). We recommend standardizing this. P4 L6: There are several places where the authors are not consistent with acronym usage (e.g. "COLD2" vs "COLD" and "MSG1" vs "MSG"). Make sure to stay consistent with these. P9 L6: Panel b of Figure 6 is never introduced in the text, so its importance is unclear. Figure 10: What is the pink region in the histogram of flight 1? That color is not in the legend. Figure 11: It is unclear why the "mean CO" black boxes represent a range. Is this supposed to be the area between the 5 and 95 percentiles? If so, a different name for this quantity should be chosen. Figure S1: Make sure to be clear in the caption that panel a is on a log scale.

Typos:

P2 L 32: Remove "for the" P4 L6: "Relative" P4 L18: "allows us to" P5 L29: "of" P6 L1: "Diabatic" in the section title P6 L24: "A higher amount of convective. . ." P10 L21: "system which developed" P12 L5: "precipitation" Figure 2: BoB is missing from the caption. Figure 7: For the description of panel e, say "below the convective cloud top." Figure 9: The caption should say that ozone is also plotted in panel c, not panel d. Figure S7: "campaign" and "27th." Table ST1: "ensemble."
* * *

---

## Author Comment (AC1) · 20 Jun 2020

**Deep convective influence on the UTLS composition in the Asian Monsoon Anticyclone region: 2017 StratoClim campaign results**

Silvia Bucci, Bernard Legras, Pasquale Sellitto, Francesco D'Amato, Silvia Viciani, Alessio Montori, Antonio Chiarugi, Fabrizio Ravegnani, Alexey Ulanovsky, Francesco Cairo, and Fred Stroh

**Answer to reviewer 1**

We want to thank the reviewer for the precious comments. We will answer point to point to them in the following:

*In this manuscript, the convective source is identified when the trajectory is found with a pressure higher than the high cloud top pressure, and the highest and opaque cloud classes are representative of the deep convective events. The features of cloud top height will determine how the convective contribution changes with height, as discussed in the later part of this manuscript. So whether the cloud top heights of these highest and opaque cloud classes can represent the deep convective main outflow levels is a critical issue. I'm wondering the uncertainty in this issue. The manuscript should have some detailed description about the uncertainty*

The work done in preparation of this paper included an extensive sensitivity study on the cloud top retrieval, as well as the choices of the Cloud Type classes as identified by the SAF algorithm. We decided not to report this whole part to avoid loading the paper too much. We agree nevertheless that this should be mentioned in the manuscript, as also the second reviewer pointed out.  We report here for representativeness an example of this sensitivity study. The SAFNWC cloud top is an operational EUMETSAT product which has been extensively validated (see for example Sèze et al. 2015). One common aspect that is often raised is the negative bias, of around 1 km, of the cloud top from geostationary with respect to LiDAR measurements (see for example Sherwood et al 2004, Haman et al 2014). We therefore repeated the analysis adding a shift of 1 km to the cloud top from the CTTH product.
We show here the result of the analysis for the same flight shown in the main text (flight 6):

[Figure]

*Figure 1A: as in figure 7c but with a +1km correction to the CTTH from the SAF product*

Comparing with figure 7c, it is possible to notice that the analysis shows a higher presence of convective influence, younger age and higher level of convective injection (this last comes by construction). The enhancement of CO are now associated to higher amounts of convective air from Pen (for the points close to segments 4 and 8) and from South-China (for the enhancement close to segment 5 and slightly before 8). Looking more in details into the source distribution (Figure2A) it appears that, in the +1Km corrected cloud top algorithm, the convective source regions (like Myanmar or South China) are associated to a larger

amount of encountered trajectories, as expected. If we now give a look to the reconstructed CO concentration (figure 3A), we see how the increase in the altitude of the clouds leads to the loss of the capture of the pollution injections (right after seconds 32000 and before 36000) that are instead seen in the non-corrected version. The trajectories are indeed "captured" as in a convective source too early in their travel back in time, marking as convective source a region that is instead not carrying the CO enriched air. The correction would give therefore more weight to clouds that are not actually influencing the observed contribution. We repeated the same study on all the fights and, while in some cases there are no remarkable changes in the results (it happens especially when there is no intense convective influence captured by the flight), when those changes are visible they appears as a degradation in the consistence of the results, as in the shown case.

A good estimation of the cloud top height is necessary for the correct reproduction of the transport in the UTLS. Using the measurements from the StratoClim campaign, we found out that the method is giving the best results when using the cloud top altitudes from the SAF without any adjustments. Physically speaking, this is also not surprising, since in this analysis we are looking for the altitude where the detrainment from the convective cloud is large enough to generate a dominating influence in the environment, that does not corresponds to the highest point of the cloud top, as it may be detected from a LiDAR profile. A more accurate specification of this notion would certainly be useful but is beyond the scope of this study.

[Figure]

*Figure 2A:  Figure 7b  (left) compared to the convective source distribution for flight 6 (as in figure 7b) but with the +1km correction to the cloud top height.*

[Figure]

*Figure 3A: as in figure 5e (but diabatic only) with (black) and without (red) a +1km correction to the cloud top height. The grey line represents the observations from COLD2*

We therefore added the following discussion to the manuscript:

"In this study we exploited the availability of the in-situ measurements of trace gases to tune for the best use of the geostationary-retrieved cloud top. While some studies indicate generally a negative bias for the geostationary-retrieved cloud top with respect to the LiDAR-retrieved one (Hamann et al. 2014), no correction to the CTTH altitude from the SAF product appears needed for the presented analysis. A sensitivity study has been performed adding different positive biases to the cloud top altitudes from the CTTH (not shown), indicating that a correction would lead to a misplacement in the identification the convective sources. A consistent interpretation of the observed tracer enhancements is instead obtained when keeping the cloud top altitudes from the SAF without any shift. The altitude on which we are interested in this study is that at which the detrainment of convective cloud is large enough to dominate the environment and the cloud top from the geostationaries may indeed be more representative of this level rather than the optical top of the cloud as it could be seen from a LiDAR profile. The selection of the cloud types to be included in the study has also been based on the same measurements-based comparison."

*The convective sources diagnosed in this manuscript are based on the StratoClim flight tracks, which only cover a small part of the Asian monsoon anticyclone, and to the south of the anticyclone center. Whether the convective sources diagnosed here are different from other regions and can represent the whole anticyclone is a critical issue, which should be considered with caution. The manuscript should discuss this issue*

The reviewer rises an important point, related to the representativeness of the convective influence captured during the campaign. The source regions that are participating the most to the composition of the air masses observed during the campaign may indeed not be the most influencing when looking to the

whole AMA composition. In fact, it is instead true that the convective events captured during the flights were strongly related to the position of the flights (mostly toward the south-center of the anticyclonic circulation) as well as to the period in which the campaign took place (including a break phase of the monsoon). While it's not the scope of the paper to give an assessment of the sources of convective activity in the whole AMA, this work provides an assessment on the quality of the approach that will be indeed exploited for a more extended analysis on the deep convective impact in the Asian Monsoon region. The question of the whole AMA source has been addressed in a companion paper (Legras and Bucci, ACPD, 2019). We therefore added the following modifications in the "discussion and conclusion" section:

"It is important to point out that the convective sources influences observed during the campaign are strongly related to the position and time period of the campaign itself. The region spanned by the aircraft is limited to the central-southern part of the anticyclone and the sources observed are therefore mainly the ones that are found upwind of the anticyclonic circulation with respect to the aircraft position. Similarly, a sampling region located more South would have likely captured more maritime convection (that is expected to be dominant with respect to the continental contribution for extension and frequency) and an additional prolongation of the campaign period after the break phase would have probably allowed to sample more intense convective events. This analysis nevertheless provided an in-situ measurement assessment of the combined satellite-modelling approach to represent the convective transport in the region, providing a tool for a reliable analysis on a longer period and a wider region. On the other hand, the analysis of the StratoClim flights provided evidence on how the convective events over these regions, even if very short-lasting and localized, may be intense enough to allow a fast and direct injection of highly polluted air at the UTLS level, spanning a large domain, that can then keep rising to enter the stratospheric circulation. As the campaign was conducted in a phase of weak convective activity and having spanned only a limited region of the AMA, it is reasonable to expect the occurrence of more intense and frequent events of fresh and eventually polluted air into the UTLS along the whole season. Those events may in principle strongly impact the chemical composition of the stratosphere. Some of these intense convective injections, while bringing polluted air at very high levels (peaks detected at 17.7 \unit{km} up to 50 \unit{ppbv} over the background) play also a role in the hydration-dehydration of the stratosphere. The discussion of the hydration effects, based on the analysis of the more resolved visible images, and the analysis on the frequency and seasonal relative impact of deep convection on wider time/space scales, are matter of ongoing investigation."

*Page 2 Line 7: This statement is valid in the upper troposphere and lower stratosphere.*

We added this information in the sentence.

*Page 2-3: These papers (Chen et al., 2012; Bergman et al., 2013; Vogel et al., 2015) have different targeted parcels (within the AMA or not) and therefore different sources, which should be noted.*

That is correct, we now mention it in the text.

*Page 3 Line 8: In situ balloon soundings over the Tibet Plateau by Bian et al. (GRL, 2012) could be mentioned here.*

Reference added.

*Page 6 Lines 19-30: The comparison among EAD, EAZ, EID and EIZ is also done by Li et al. (ACPD, 2019), which could be cited here*

Reference added

*Page 7 Line 5-6: The plume is located at #6 at 34000s UTC?*

No, we refer here to the two enhancements right after seconds 32000 and before 36000s. We specified this in the text.

*Page 9 Line 5: F6 therefore sampled the inner part of the AMA. Whether the parcels sampled during F6 are called "inner part" depend how the inner part of the AMA is defined. The 100 hPa circulation is centered around 33N, while the flight track position is 20-26 N, where is 7-13 degrees from the center. China sources make a critical contribution to the south part of AMA, which has also been shown by Yan & Bian (AAS, 2015)*

Inner side here is used not with the meaning of "centre" but rather to indicate that, with respect to the geopotential contours, we are not on the outer side of the anticyclonic circulation, whatever metric is used. Indeed, in the case of flight 6 (figure S7 panel f) the circulation of the AMA was such that the Chinese sources were not just on the south part of the AMA (which fluctuates a lot) but are ventilated by the streamlines close to the centre of the circulation.

*Page 9 Line 6: The cold point pressure is derived from ERA5?*

Yes, we added the indication in the text.

*Page 9 Line 7-8: flight. The mean winds around the flight position were purely Easterlies, transporting air from the center of South China along the anticyclonic circulation, which is also the main reason why South China contributes a lot in the convective sources as discussed in Page 7. This issue should be mentioned there.*

We agree on the importance to mention that the relative importance of the sources is dependent on the position of the flight. We therefore added the discussion of this aspect in the "conclusion and discussion" section.

*Page 9 Line 11-12: I'm curious if there were no aircraft dive here, would the CO peak be observed?*

The dependence of the CO concentration on the altitude has been taken in account subtracting the average CO profile. The variations in CO concentration shown in figure 5 are therefore not consequence of the variation of the flight level but to the presence of the convective plume. The dive may have had nevertheless a role in the sense that it may have increased the possibility of encounter of convective air by sampling a lower altitude where the chance of convective presence is larger. We added this discussion in the text.

Reference:

Hamann, U., Walther, A., Baum, B., Bennartz, R., Bugliaro, L., Derrien, M., Francis, P. N., Heidinger, A., Joro, S., Kniffka, A., Le Gléau, H., Lockhoff, M., Lutz, H.-J., Meirink, J. F., Minnis, P., Palikonda, R., Roebeling, R., Thoss, A., Platnick, S., Watts, P., and Wind, G.: Remote sensing of cloud top pressure/height from SEVIRI: analysis of ten current retrieval algorithms, Atmos. Meas. Tech., 7, 2839–2867, https://doi.org/10.5194/amt-7-2839-2014, 2014

Sèze, G., Pelon, J., Derrien, M., Le Gléau, H. and Six, B. (2015), Evaluation against CALIPSO lidar observations of the multi-geostationary cloud cover and type dataset assembled in the framework of the Megha-Tropiques mission. Q.J.R. Meteorol. Soc., 141: 774-797. doi:10.1002/qj.2392

Sherwood, S. C., Minnis, P., and McGill, M. ( 2004), Deep convective cloud-top heights and their thermodynamic control during CRYSTAL-FACE, J. Geophys. Res., 109, D20119, doi:10.1029/2004JD004811

---

## Author Comment (AC2) · 20 Jun 2020

**Deep convective influence on the UTLS composition in the Asian Monsoon Anticyclone region: 2017 StratoClim campaign results**

Silvia Bucci, Bernard Legras, Pasquale Sellitto, Francesco D'Amato, Silvia Viciani, Alessio Montori, Antonio Chiarugi, Fabrizio Ravegnani, Alexey Ulanovsky, Francesco Cairo, and Fred Stroh

**Answer to reviewer 2**

We thank the reviewer for the insightful comments. We will answer point to point to them in the following:

*Scientifically, the manuscript does not adequately define the objectives of the study in the scope of the outstanding scientific questions. As a result, it is not clear what the key conclusions are. The only clear conclusion is related to the better performance of ERA-5 and diabatic vertical wind as opposed to ERA-Interim and kinematic wind. This conclusion does not support the title of the article.*

We reformulated the objective of the paper in the abstract and the introduction. Indeed, the manuscript is intended to give the first in situ measurement-supported analysis of the convective origin for the air masses close to the tropical tropopause in the Asian monsoon region. As the StratoClim airborne campaign is the first ever conducted in this area, this is by itself an important result to report. Among the outstanding results there is the evidence of fresh pollution injection directly below the tropopause, the detection of the intense overshoots at the level of the tropopause, as well as the presence of typhoon injected air. The strength of this paper is based on the combination of a modelling approach that is tuned, and at the same time supported, by the in-situ observations. It is the first time that such comparison can be made in the AMA.

*The lack of a clear objective is also reflected in the description of the depth of convective "injection" and the "age" of air in the samples. First of all, no clear definition is given to define "age". We assume it is defined by the length of the back-trajectory from flight track to convective top encounter, but what is the physical significance of this quantity?*

We also better present those two quantities. The depth of convective injection is given by the cloud top pressure, that we suppose to be close to main level of convective detrainment. This is an approximation that takes in account that the cloud top identified by the SAF algorithm, which is that of main radiative emissivity, is not the optical highest point of the cloud. The age instead is indeed the time between the detrainement from the cloud (assumed to be instantaneous and coinciding with the first encounter of the clouds by the trajectories) and the moment of the observation (corresponding to the trajectories release), representing the life time of air in the UTLS which is highly relevant for chemical processes. Physically we can assume that this time is pretty close to the real time of transport from the boundary layer, since what is missing is the estimate of the time of vertical convective transport from the boundary layer to the cloud top. As the sustained vertical velocity of the updrafts inside convective towers is typically of the order of 10 m/s, the time span of upflit from the boundary layer is of few hours at most. We can therefore ignore this delay. The analysed vertical velocities used in many studies are much weaker but they are only representative of the mean motion over cells much wider than the convective updrafts.

*Revise the introduction to clearly state the objectives. If identifying the chemical characteristics of convectively transported air from different regions is the ultimate goal of the data analysis, but the composition analysis is beyond this work alone, it still needs to clearly articulated. The regions used for "airmass origin apportionment" should also be defined with different chemical emission characteristics or convection behaviors in mind*

Following the reviewer suggestion, we fixed the introduction to clarify this point. The reviewer raises the importance of the choice of the airmass origin apportionment. Indeed, for this work we looked for a good definition of the convective / most emissive regions. Nevertheless, the study has as a first broader objective to present a tool for the convective analysis in the UTLS that could be applied to different atmospheric components which path of injection is not always relatable to the CO emissions. Similarly, to define the regions based only on the convective occurrence, would be to limit the selection or to the campaign-related main source or to the more general convective activity frequency (that is differently distributed). Since the study wants to be an opener for more comprehensive analysis of seasonal convective injection while still relying on the campaign results, we decided to keep a more neutral division of the regions mainly based on the country borders. Country borders anyway are related to geographical natural borders (mountains, rivers, land-ocean separation) as well as being related to differences in economy and policies of anthropogenic emissions. The inclusion of the Tibetan Plateau region and the separation of the South China region from the North one has been intended indeed to take into account that those two regions are peculiar under the point of view of frequency of occurrence and/or in pollutants emissions. A more detailed identification of the sources can still be identified in the source distribution plots (as in panel a) of figures 7,9,S1,S2,S3,S4,S5,S6).

*An implicit goal of the work is to investigate how often and how deep the convective transport is influencing the "UTLS" composition. Separating the UT from the LS is important. If the tropopause identification is not supported by the flights themselves, an estimate using ERA5 data could still be very helpful to quantify and characterize the height of convective "injection" relative to the tropopause. With the help of the tropopause location, quantifying the direct influence of convection relative to the tropopause, the relative contribution from the regions in the UT, and when and where convection influenced the stratosphere can be a significant conclusion of this work.*

As the review points out, the intention of the paper is to emphasize the results on the influence of deep convective transport onto the UTLS composition, so we decided to reinforce this concept in the introduction. The tropopause has been identified as a matter of fact by the flight measurements when extensive profiles were available, otherwise this is identified by ERA5 indeed, as shown in figures 6 and 8 panels a. ERA5 was recently demonstrated to have the best tropopause height estimate among the reanalysis (Tegtmeier et al. 2020). We therefore emphasized this better in the text. Furthermore, it is demonstrated in a companion paper (Legras and Bucci, ACPD, 2019) that the tropical tropopause is not associated to any discontinuity in the transport properties at the scale of the AMA when they are seen in the potential temperature framework. Therefore, if it remains important to localize the convection tops with respect to the tropopause, the transition from UT to LS is actually very smooth regarding transport properties.

*From the presentation point of view, the manuscript suffers from a deficiency of too many details and lack of clear take home message. Although there are many details highlighting convective influence from different boundary layers (such as Northern India and the Tibetan plateau), there is no clear message why transport from these regions is important. For the two selected flights described segment by segment, the writing style is similar to that of a detailed flight report. Convective origin, or a sample's "airmass source apportionment", is a big focus of the analysis, but no significant chemical consequences are shown from the analysis or articulated in the introduction. After all these details, it is not clear what the significant findings*

*are or what is scientifically new. We suggest that the discussion and analysis be reorganized around new findings.*

As already stated, we are assessing the transport properties in the AMA based on the first ever high altitude airborne campaign in the region. This is by itself very new. We worked on the text to re-focus the conclusions on the main objective: the identification and quantification of the percentage and age of tropospheric air injected by deep convection into the UTLS, supported by in-situ observations. Moreover we clarify that giving a description of the chemical consequences is out of the scope of the paper, that is meant to focus on the purely dynamical aspect of the transport (using CO not for chemical study purposes but as a tropospheric transport tracer since it is an indicator of anthropogenic influenced air). On the other hand, this study is also meant to be a reference for transport characterization for the other StratoClim papers, that will be indeed aiming to describe more in details the chemical aspects. For those reasons we strengthened the objectives description while preserving the needed detailed segment by segment analysis as a reference for the future papers.

*A number of sections are written as one paragraph. It seems largely due to the style of "flight logging" used throughout. This poses a challenge for the readers. We suggest that the authors highlight the main goal of each discussion, select significant details, and break the sections into a number of paragraphs according to the take-home messages.*

We took this suggestion in account in the review of the main text, we therefore added the highlights of the main results of the flight analysis and restructured some of the paragraphs.

*All the key information would need to be in the paper, not the supplement. For example, it is important to show the flight tracks relative to the flow pattern of the anticyclone*

Since we want to preserve the details on the convective influence and the tropopause variability shown in the closer look of figures 6 and 8, but without expanding the main paper with further figures, we would still prefer to keep the wider circulation on the supplement. We agree nevertheless that it would be useful to have a view of the flight position with respect to the anticyclonic circulation and decided therefore to add the flight tracks to the panels of figure S7. Notice also that such general figures will be found in a forthcoming overview paper that will cap the series of papers in the StratoClim special issue of ACP.

*It is also a good practice to make the figures, including the titles and axis labels, large enough to read in the printed version. There are a number of issues with this, including Figures 3, 6 and 8.*

Following the reviewer advice, we enlarged the fonts of the figures.

*The large number of regions defined in Figure 3 should be re-considered since the authors seem to have run out of colors to represent all the regions distinctly. For example, in the later figures, Tibetan plateau, MPac, Bangladesh and Pakistan are not always separable, especially in the print version.*

To take into account the reviewer remark we remove the Japanese-Korean and the North-China regions, that are not significant convective contributors with respect to the other regions, and changed the colors of MPac and Bangladesh and Tibetan Plateau.

**Specific Comments**

*We suggest that the authors address the issue of CO being the only chemical convectively influenced composition variable shown in the manuscript, since the paper is about the convective influence on the UTLS composition. We note that the use of reconstructed CO to diagnose the trajectory based convective transport identification is a nice piece of analysis in this work. CO alone, however, doesn't represent the objective of the campaign. It would be good to state the limited objective of using CO in this analysis and*

*the goal of the analysis is to support the full scope of chemical composition analysis, in particular the short lived active species, etc.*

The comment of the reviewer raised the need to clarify in the text the choice of the use of CO and that we focus on the transport properties rather than on the chemistry. As mentioned above, the CO mixing ratio is used here as an indicator for the tropospheric air presence, as it is a good tracer for anthropogenic pollution, with a lifetime in the atmosphere of around 2 months, compatible with our trajectories time. Therefore the scope is not to have a full chemical composition analysis but to give a reliable deep convective transport description backed up by the observations. This has been clarified in the text.

*A statement about the magnitude of uncertainty in satellite-derived convective cloud tops would be beneficial since the results hinge very strongly on these being accurate.*

Following the advices of both reviewers, we included a statement on the sensitivity study we performed with different cloud top altitudes corrections. See also the answer to the first reviewer.

*If the assumption is made that once a parcel encounters a convective cloud top it is considered to simultaneously contact the boundary layer, it is an important assumption to be explicitly stated and justified.*

When the parcel encounters a convective cloud top is considered to have been instantaneously detrained from there, while the time from the departure from the boundary layer can be estimated to be of the order of few hours but is not taken into account in the computations. This is now stated in the text, see also previous answers. One of the main point of this work is indeed that we do not use the vertical winds from the reanalysis for the convective transport as they are not representative of convective transport and result in unrealistic time of transport from the boundary layers (weeks instead of hours).

*P11 L12-14: It is an inaccurate statement of "stratospheric intrusion" based on the observed CO-O3 structure without a tropopause analysis. It is also possible the flight sampled a filament of stratospheric air produced by the large scale stirring.*

We simply mean indeed that it is a sample of stratospheric air, we correct this in the paper.

*P4 L23-25: "The trajectories move . . ." needs to be revised. This sentence has no clear meaning. Do the authors intend to say "Only the trajectories moving within the domain 10-160 E and 0-50N are considered" in the analysis?*

We intend to say that the trajectories are bounded to the limited domain of the meteorological fields (cutted to 10-160E and 0-50N for ERA5 for computational reasons), therefore they cannot be transported outside those boundaries since there would be no wind fields there. In this case the trajectories are considered there to be "dead". This fraction of "dead" parcels corresponds to the white space in the contributors' percentage plot. We clarify this point in the text.

*There is an inconsistency between Sections 1 and 2 about when StratoClim ended (beginning or middle of August). We recommend standardizing this*

We corrected this, the campaign ended on 10[th] of August so we decided to stick with "middle of August"

*P4 L6: There are several places where the authors are not consistent with acronym usage (e.g. "COLD2" vs "COLD" and "MSG1" vs "MSG"). Make sure to stay consistent with these*

We thank the reviewer for mentioning this, we fixed it in the text.

*P9 L6: Panel b of Figure 6 is never introduced in the text, so its importance is unclear.*

The image is meant to show the convective situation in the vicinity of the flight. Is possible to see there how the CO enhancement in the flight are not due to transport from close convective systems upwind, as well as showing that there is no overpass of the flight on convective systems (as instead happens in fight 8). We now mention it in the manuscript.

*Figure 10: What is the pink region in the histogram of flight 1? That color is not in the legend.*

It was the Japanese region, now removed according to the previous comments. We therefore updated the figure.

*Figure 11: It is unclear why the "mean CO" black boxes represent a range. Is this supposed to be the area between the 5 and 95 percentiles? If so, a different name for this quantity should be chosen*

The use of bars in the figure may indeed be misleading, since the values represented are not ranges but simple means of the CO anomaly (black), 5 percentile (light grey) and 95 percentile (dark grey). We therefore substituted it with a line plot.

*Figure S1: Make sure to be clear in the caption that panel a is on a log scale.*

Correct, we fixed the caption.

*Typos: P2 L 32: Remove "for the" P4 L6: "Relative" P4 L18: "allows us to" P5 L29: "of" P6 L1: "Diabatic" in the section title P6 L24: "A higher amount of convective. . ." P10 L21: "system which developed" P12 L5: "precipitation" Figure 2: BoB is missing from the caption. Figure 7: For the description of panel e, say "below the convective cloud top." Figure 9: The caption should say that ozone is also plotted in panel c, not panel d. Figure S7: "campaign" and "27th." Table ST1: "ensemble."*

We corrected the typos, we thank the reviewer for identifying them.

References:

Legras, B. and Bucci, S.: Confinement of air in the Asian monsoon anticyclone and pathways of convective air to the stratosphere during summer season, Atmos. Chem. Phys. Discuss., https://doi.org/10.5194/acp-2019-1075, revised sub judice, 2019.

Tegtmeier, S., Anstey, J., Davis, S., Dragani, R., Harada, Y., Ivanciu, I., Pilch Kedzierski, R., Krüger, K., Legras, B., Long, C., Wang, J. S., Wargan, K., and Wright, J. S.: Temperature and tropopause characteristics from reanalyses data in the tropical tropopause layer, Atmos. Chem. Phys., 20, 753–770, https://doi.org/10.5194/acp-20-753-2020, 2020.